# NeUQI: Near-Optimal Uniform Quantization Parameter Initialization for Low-Bit LLMs

**Li Lin** [1]  **Xinyu Hu** [1]  **Xiaojun Wan** [1]

## Abstract

Large language models (LLMs) achieve impressive performance across domains but face significant challenges when deployed on consumer-grade GPUs or personal devices such as laptops, due to high memory consumption and inference costs. Post-training quantization (PTQ) of LLMs offers a promising solution that reduces their memory footprint and decoding latency. In practice, PTQ with uniform quantization representation is favored due to its efficiency and ease of deployment, as uniform quantization is widely supported by mainstream hardware and software libraries. Recent studies on low-bit uniform quantization have led to noticeable improvements in post-quantization model performance; however, they mainly focus on quantization methodologies, while the initialization of quantization parameters remains underexplored and still relies on the conventional *Min-Max formula*. In this work, we identify the limitations of the *Min-Max formula*, move beyond its constraints, and propose **NeUQI**, a method that efficiently determines near-optimal initialization for uniform quantization. Our NeUQI simplifies the joint optimization of the scale and zero-point by deriving the zero-point for a given scale, thereby reducing the problem to a scale-only optimization. Benefiting from the improved quantization parameters, our NeUQI consistently outperforms existing methods in the experiments with the LLaMA and Qwen families on various settings and tasks. Furthermore, when combined with a lightweight distillation strategy, NeUQI even achieves superior performance to PV-tuning, a considerably more resource-intensive method.

[1]Wangxuan Institute of Computer Technology, Peking University, China. Correspondence to: Xiaojun Wan <wanxiaojun@pku.edu.cn>.

*Proceedings of the $43^{rd}$ International Conference on Machine Learning*, Seoul, South Korea. PMLR 306, 2026. Copyright 2026 by the author(s).

## 1. Introduction

In recent years, large language models (LLMs) like Chat-GPT (OpenAI et al., 2024) have rapidly emerged, demonstrating strong capabilities across various tasks, including open-ended writing, knowledge-based question answering, and code generation. Given the high API costs of proprietary LLMs and the growing performance of open-source alternatives like LLaMA (Touvron et al., 2023; Grattafiori et al., 2024) and Qwen (Yang et al., 2025) families, there is a rising preference for deploying open-source LLMs locally. However, the deployment of large-scale models (e.g., LLaMA 3 70B) is often limited by compute resources and inference efficiency, especially on personal devices or consumer-grade GPUs like a single RTX 4090 with 24GB of memory. In this context, post-training quantization (PTQ) (Krishnamoorthi, 2018), particularly using a uniform quantization scheme, offers a practical solution by converting model weights from `bfloat16` to low-bit-width integer formats such as `int4/3/2`, significantly reducing memory usage and inference latency.

In the context of uniform quantization, the Min-Max initialization method (Jacob et al., 2017) is simple and effective at higher bit-widths (e.g., 8-bit, 4-bit) (Dettmers et al., 2022; Frantar et al., 2023), but its simplistic design becomes ineffective in lower bit-width settings such as 2-bit and 3-bit. Although prior works on uniform PTQ have achieved notable progress, they still fundamentally rely on the *Min-Max formula* for initialization. This formulation imposes two key constraints on the optimization process, which have been largely overlooked and give rise to practical limitations, as discussed in Section 3.3: (1) both the scale and the zero-point are determined by extreme values, thereby constraining the flexibility of the optimization algorithm; and (2) the zero-point is restricted to be an integer, which limits the parameter space. The drawback of the first constraint is most pronounced in its unnecessary enlargement of the search space for search-based methods such as Lean-Quant (Zhang & Shrivastava, 2025).

After recognizing the two constraints inherent in the *Min-Max formula*, we move beyond them and develop NeUQI to explicitly exploit the resulting loss structure. To address the joint optimization of the scale and zero-point, we propose

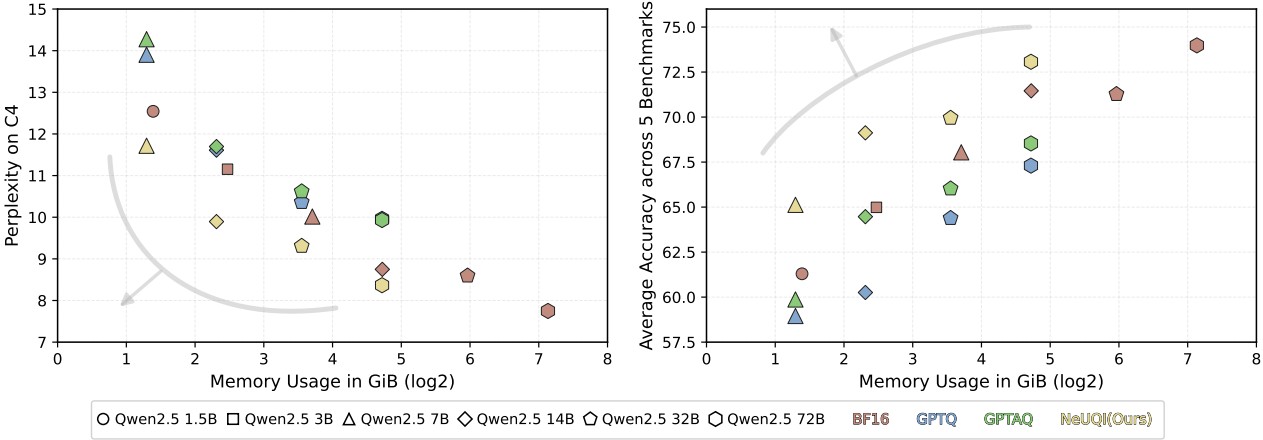

*Figure 1.* Perplexity on C4 (left) and average accuracy across five common benchmarks (right) with Qwen 2.5 family, plotted against the base-2 logarithm ($\log_2$) of model memory usage on the x-axis. Gray arrows indicate the direction of better performance, corresponding to lower perplexity and higher accuracy with smaller model memory usage. The results include non-quantized models (BF16) and quantized models at 3-bit using GPTQ, GPTAQ, as well as our proposed NeUQI. More details are provided in Section 5.

an efficient method that computes a near-optimal zero-point for a given scale, reducing the problem to a single-variable optimization over the scale, which is efficiently solved via a coarse-to-fine search. Empirical results show that our NeUQI substantially improves the performance of quantized models and, under the 3-bit channel-wise setting, can even surpass that of non-quantized models with comparable memory usage, as illustrated in Figure 1. By moving beyond the *Min-Max formula* in determining quantization parameters, NeUQI provides an effective initialization that enables lightweight distillation to surpass PV-tuning (Malinovskii et al., 2024), a more sophisticated and resource-intensive fine-tuning method, and further yields additional gains when integrated in the strong fine-tuning method EfficientQAT (Chen et al., 2025). These findings underscore the critical role of better initialization: poor initial conditions are difficult to fix even with extensive fine-tuning, whereas a well-initialized model can achieve better performance with significantly less effort.

The main contributions of our work are summarized as follows:

1. We identify two previously overlooked constraints in the commonly adopted *Min–Max formula* for uniform PTQ initialization, revealing its inherent limitations and offering new insights into initialization design.

2. Motivated by these limitations and the resulting loss structure, we propose NeUQI, a novel and efficient initialization method for uniform quantization.

3. Extensive experiments on the LLaMA and Qwen model families across various sizes and tasks demon-

strate that our NeUQI consistently outperforms existing methods. Furthermore, experiments on fine-tuning methods such as PV-tuning and EfficientQAT highlight the importance of better initialization and the effectiveness of NeUQI.

## 2. Related Work

**Uniform Quantization** In the 2-bit to 4-bit quantization regime, prior studies have explored a range of post-training quantization techniques to mitigate accuracy degradation. Early methods such as SmoothQuant (Xiao et al., 2023) and AWQ (Lin et al., 2024b) applied channel-wise transformations, shrinking activations while compensating by enlarging the corresponding weights. Subsequent works extended channel-wise scaling into efficient invertible transformations for improved smoothing distribution, as exemplified by QuIP (Chee et al., 2023), DuQuant (Lin et al., 2024a), FrameQuant (Adepu et al., 2024), and QuaRot (Ashkboos et al., 2024). Among these, the Hadamard transform, widely adopted as a representative transformation-based technique, offers simplicity while maintaining comparable effectiveness, as demonstrated in QuaRot. Alternatively, MagR (Zhang et al., 2024) suppresses the magnitude of weights while preserving model functionality. In addition, fine-tuning-based approaches have been explored, such as CBQ (Ding et al., 2025), which performs block-wise tuning, and PV-Tuning (Malinovskii et al., 2024), which jointly optimizes continuous and discrete parameters. Meanwhile, SpinQuant (Liu et al., 2025) and OSTQuant (Hu et al., 2025) incorporate learnable transformations into the model, substantially reducing runtime overhead.

**Quantization Parameter Initialization** Most uniform quantization methods use the conventional Min-Max initialization, while the rest employ variants such as clipping-based, quantile-based, MSE-based, and MeanStd-based approaches (Li et al., 2021). Among them, the MSE-based variant is search-based, whereas the others rely on empirical formulas to estimate the scale and zero-point. These variants also inherit the two key constraints of the *Min–Max formula* and their associated limitations. The fine-tuning-based method OmniQuant (Shao et al., 2024) extends clipping-based Min-Max into learnable weight clipping during training; however, it cannot leverage results obtained by non-fine-tuning methods. Although LeanQuant (Zhang & Shrivastava, 2025) determines parameters through a loss-aware search rather than directly applying Min-Max or its variants, it still suffers from the same limitation, thereby constraining the design of the optimization algorithm.

# 3. Background

## 3.1. Uniform Quantization

Quantization restricts numerical values to a finite indexed set $\mathcal{Q}$, enabling each value to be stored and communicated as a low-bit-width index, thereby reducing memory footprint and memory bandwidth cost. In the case of **uniform quantization**, more commonly known as **asymmetric affine quantization**, $\mathcal{Q}$ consists of equally spaced real values parameterized by the scale $s$, the zero-point $z$, and the bit-width $k$. Concretely,

$$\mathcal{Q} = \{q_i\}_{i=0}^{2^k-1}, \quad q_i = s \cdot (i - z). \quad (1)$$

To reduce quantization error, the quantization function typically maps an input value to the nearest element in $\mathcal{Q}$. Given the input $x$, the uniform quantization function $Q$ is defined as:

$$Q_{s,z}(x) = s \cdot \left( \text{clip}(\lfloor x/s + z \rceil, 0, 2^k - 1) - z \right), \quad (2)$$

where $\lfloor \cdot \rceil$ is the rounding operator and $\text{clip}(\cdot, a, b) = \max(a, \min(\cdot, b))$ restricts the value to $[a, b]$. When the input to $Q$ is a vector or a matrix, the quantization function is applied element-wise. Notably, conventional formulations (Krishnamoorthi, 2018; Jacob et al., 2017) add the zero-point after the rounding operator, which forces $z$ to be a $k$-bit unsigned integer. In contrast, our formulation imposes no integer constraint on $z$, and thus **allows $z$ to take floating-point values**.

## 3.2. Min-Max Initialization

Min-Max determines the quantization parameters by mapping the minimum and maximum of the parameter vector $\boldsymbol{x}$ to the minimum and maximum of quantized values. Based on the extremal-value mapping under the conventional setting, the ***Min-Max formula*** is given as follows, from which the scale $s$ and the zero-point $z$ are directly determined:

$$s = \frac{\max(\boldsymbol{x}) - \min(\boldsymbol{x})}{2^k - 1}, \quad z = -\left\lfloor \frac{\min(\boldsymbol{x})}{s} \right\rceil. \quad (3)$$

Under this formulation, the resulting value of $z$ is typically an integer within the range $[0, 2^k - 1]$, that is, a $k$-bit unsigned integer.

## 3.3. Constraints of the Min-Max formula

**The first constraint** of the *Min-Max formula* determines the scale and zero-point from fixed extreme values, which biases search-based initialization toward enumerating scaling factors associated with the minimum and maximum. In methods such as LeanQuant, this results in a two-dimensional grid search over Min-Max scaling factor pairs, with $T$ candidates for each extreme and thus $T^2$ evaluations, where $T$ is typically 2048. By contrast, searching directly in the scale and zero-point spaces only requires $T$ scale candidates and $2^k$ zero-point candidates (Jacob et al., 2017) with $k \leq 4$, resulting in $2^k T$ evaluations. Since $T^2 \gg 2^k T$, the extreme-value-based constraint tends to induce substantially slower search designs.

**The second constraint** of the *Min-Max formula* requires the zero-point to be a $k$-bit unsigned integer in $k$-bit quantization, which imposes a significant limitation by reducing the quantization parameter space and constraining achievable performance, particularly in low-bit-width settings. In contrast, NeUQI relaxes this constraint and consequently achieves superior performance, as shown in Section 5.3 and Section 5.4, with only a marginal increase in average bit-width. Table 5 further shows that NeUQI outperforms the strongest baseline that enforces integer zero-points, despite operating at a lower average bit-width. In particular, compared to the strongest baseline, for LLaMA 2 7B quantized at an average bit-width of approximately 2.25, NeUQI reduces C4 perplexity by about 15.54% and improves the average accuracy across five benchmarks by approximately 3.95%. For details on hardware support for floating-point zero-points, see Appendix E.

# 4. Methodology

In this section, we present **NeUQI** (**Ne**ar-Optimal **U**niform **Q**uantization **I**nitialization), a novel method designed to improve uniform quantization initialization. Our NeUQI moves beyond the two constraints of the *Min-Max formula*, enabling more effective direct optimization of the scale and zero-point. In the following, we first revisit the formulation of quantization loss, then explain how NeUQI performs the optimization.

## 4.1. Quantization Loss

Following the formulation adopted in GPTQ (Frantar et al., 2023), the quantization loss for a single linear layer is defined as

$$
\begin{aligned}
\mathcal{L}(\boldsymbol{W}, s, z) &= \mathbb{E}_{\boldsymbol{X}}[\|\boldsymbol{X}(Q_{s,z}(\boldsymbol{W}) - \boldsymbol{W})^{\top}\|_F^2] \\
&= \text{tr}\left((Q_{s,z}(\boldsymbol{W}) - \boldsymbol{W})\boldsymbol{H}(Q_{s,z}(\boldsymbol{W}) - \boldsymbol{W})^{\top}\right) \\
&= \sum_i (Q_{s,z}(\boldsymbol{W}_{i,:}) - \boldsymbol{W}_{i,:})\boldsymbol{H}(Q_{s,z}(\boldsymbol{W}_{i,:}) - \boldsymbol{W}_{i,:})^{\top},
\end{aligned}
\tag{4}
$$

where $\boldsymbol{X}$ denotes the input activations, $\boldsymbol{W}$ is the weight matrix of the linear layer. $\boldsymbol{H} = \mathbb{E}_{\boldsymbol{X}}[\boldsymbol{X}^{\top}\boldsymbol{X}]$ is commonly referred to as the proxy Hessian of $\boldsymbol{W}_{i,:}$. This formulation naturally suggests adopting the row-wise quantization strategy, which aligns with common practices that treat one row as one or multiple parameter vectors for quantization.

## 4.2. NeUQI

Although the full Hessian matrix can be efficiently computed for a single linear layer, incorporating it into optimization is often hard to handle in practice. To address this, we adopt the diagonal approximation (LeCun et al., 1989; Liu et al., 2024), a widely used and practical approximation that treats cross-weight interactions as negligible and yields a simpler loss function while maintaining global coupling through shared quantization parameters. Under this approximation, the loss that our method aims to minimize for initializing the quantization parameters of one row $\boldsymbol{w}$ reduces to:

$$
\mathcal{L}(s, z) = \sum_i H_{i,i}(Q_{s,z}(w_i) - w_i)^2.
\tag{5}
$$

To address the coupled optimization of the scale and zero-point, we first fix the scale and propose a method that efficiently computes a near-optimal zero-point. This reduces the original two-variable problem to a single-variable optimization over the scale, which is solved using a coarse-to-fine search strategy to reduce computation time while maintaining accuracy.

### 4.2.1. OPTIMIZATION OF ZERO-POINT

Fixing the scale, with $x_i = w_i/s$ and $h_i = H_{i,i}s^2$, the loss function becomes

$$
\mathcal{L}(z) = \sum_{i=1}^{n} h_i\big(x_i + z - \text{clip}(\lfloor x_i + z \rceil, 0, 2^k - 1)\big)^2.
\tag{6}
$$

---

**Algorithm 1** Optimal Zero-point over Piecewise Quadratic Function

**Input** Piecewise quadratic function $\{\mathcal{L}_i(z)\}_{i=1}^n$
Initialize list of transition points: $\mathcal{T} \leftarrow [\,]$
**for** each piecewise quadratic function $\mathcal{L}_i(z)$ **do**
    **for** each transition point index $j$ on $\mathcal{L}_i(z)$ **do**
        $t \leftarrow$ the $j$-th transition point of $\mathcal{L}_i(z)$
        $\delta(z) \leftarrow$ the $(j+1)$-th interval function of $\mathcal{L}_i(z)$ subtracted by the $j$-th interval function
        Add $(t, \delta(z))$ to $\mathcal{T}$
    **end for**
**end for**
Sort $\mathcal{T}$ by transition point $t$
$\mathcal{L}^{\text{I}}(z) \leftarrow$ sum of the first intervals of all $\mathcal{L}_i(z)$
$(t_{\text{first}}, \delta_{\text{first}}) \leftarrow$ first element of $\mathcal{T}$
$z' \leftarrow \arg\min_{z \in (-\infty, t_{\text{first}}]} \mathcal{L}^{\text{I}}(z)$
$(z^*, \mathcal{L}^*) \leftarrow (z', \mathcal{L}^{\text{I}}(z'))$
**for** each $(t, \delta(z))$ in $\mathcal{T}$ **do**
    $\mathcal{L}^{\text{I}}(z) \leftarrow \mathcal{L}^{\text{I}}(z) + \delta(z)$ {Transition from the previous interval to the current one via the increment.}
    Let next transition point be $t'$ (or $+\infty$ if none)
    $z' \leftarrow \arg\min_{z \in [t, t']} \mathcal{L}^{\text{I}}(z)$
    update $(z^*, \mathcal{L}^*)$ with $(z', \mathcal{L}^{\text{I}}(z'))$ if $\mathcal{L}^{\text{I}}(z') < \mathcal{L}^*$
**end for**
**Return** $z^*$

---

For sample $i$, the per-sample loss function is

$$
\begin{aligned}
\mathcal{L}_i(z) &= h_i\big(x_i + z - \text{clip}(\lfloor x_i + z \rceil, 0, 2^k - 1)\big)^2 \\
&= \begin{cases}
h_i(x_i + z - 0)^2, & z < \frac{1}{2} - x_i, \\
h_i(x_i + z - j)^2, & j - \frac{1}{2} - x_i \le z < j + \frac{1}{2} - x_i, \\
& \qquad j = 1, \ldots, 2^k - 2 \\
h_i(x_i + z - (2^k - 1))^2, & z \ge 2^k - \frac{3}{2} - x_i.
\end{cases}
\end{aligned}
\tag{7}
$$

Hence, $\mathcal{L}_i(z)$ and $\mathcal{L}(z)$ are piecewise quadratic functions of $z$. To determine the global minimum of $\mathcal{L}(z)$, it is necessary to obtain the explicit quadratic function on each interval. As shown in Eq. 7, $\mathcal{L}_i(z)$ has $2^k - 1$ transition points and $2^k$ intervals. The transition points of $\mathcal{L}(z)$ are the union of those from all $\mathcal{L}_i(z)$. For analytical and computational purposes, coincident transition points are treated as distinct by inserting zero-length intervals, thereby preserving the function values and leaving the outcome unchanged. As a result, $\mathcal{L}(z)$ has $n(2^k - 1)$ transition points and $n(2^k - 1) + 1$ intervals. A naive approach that iterates over all intervals and, for each interval, accumulates the corresponding interval functions $\mathcal{L}_i(z)$ over all $i$, resulting in a time complexity of $O(n2^k \cdot n)$, which becomes prohibitively expensive in practice when $n$ is on the order of 2048.

Next, we observe that between two adjacent intervals of $\mathcal{L}(z)$, only one $\mathcal{L}_i(z)$ changes. Consequently, the change in the quadratic function between two adjacent intervals of $\mathcal{L}(z)$ is exactly equal to the corresponding change between the two adjacent intervals of that $\mathcal{L}_i(z)$. When the intervals

are enumerated in increasing order, the quadratic function on the current interval can therefore be obtained by updating the resulting quadratic function from the previous interval using the corresponding function change, rather than recomputing the full sum from scratch. As a result, the overall complexity of obtaining the quadratic function on every interval of $\mathcal{L}(z)$ is reduced to $O(n2^k \log(n2^k))$, where sorting all transition points dominates the total running time. The high-level procedure is summarized in Algo. 1 applied to Eq. 7, and the complete algorithm is provided in Algo. 2 in Appendix C.

However, even under the most common settings of $k = 2, 3, 4$, the resulting computational burden remains substantial. Since the complexity of the above algorithm is dominated by the number of transition points, we adopt the following strategy to reduce this number, thereby achieving an overall time complexity of $O(n \log n)$. Specifically, in Eq. 7, we assume that $\mathcal{L}_i(z)$ reaches its upper bound of $h_i \cdot \frac{1}{4}$ over the low-loss interval $[-\frac{1}{2} - x_i, 2^k - \frac{1}{2} - x_i]$, while the remaining terms are left unchanged. Under this assumption, Eq. 7 takes the following form:

$$
\mathcal{L}_i^S(z) = \begin{cases} h_i(x_i + z - 0)^2, & z < -\frac{1}{2} - x_i, \\ h_i \cdot \frac{1}{4}, & -\frac{1}{2} - x_i \le z < 2^k - \frac{1}{2} - x_i, \\ h_i(x_i + z - (2^k - 1))^2, & z \ge 2^k - \frac{1}{2} - x_i. \end{cases}
\tag{8}
$$

Under this approximation, each sample yields exactly two transition points, located at $-\frac{1}{2} - x_i$ and $2^k - \frac{1}{2} - x_i$. By applying Alg. 1 to Eq. 8, we obtain the optimal zero-point $z^S$ of Eq. 8 in $O(n \log n)$ time. We then assume that the true optimum lies in the neighborhood of $z^S$ and apply Alg. 1 to Eq. 7 within the restricted interval $[z^S - 1, z^S + 1]$. Since the transition points defined in Eq. 7 are spaced by 1, at most two transition points per sample lie in this interval when the right endpoint is excluded, and thus this step also runs in $O(n \log n)$ time. The detailed algorithms are provided in Alg. 3 and Alg. 4 in Appendix C.

### 4.2.2. OPTIMIZATION OF SCALE

As shown in the previous section, fixing the scale enables an explicit solution for $z$, thereby reducing the loss function in Eq. 5 to a single-variable function of $s$. This formulation motivates a search-based strategy for scale selection.

To define the search space, we take the scale derived from the *Min–Max formula* as the upper bound and uniformly sample $T$ candidate scales from 0 to this upper bound:

$$
\mathcal{S}_T = \left\{ \frac{\max(\boldsymbol{w}) - \min(\boldsymbol{w})}{2^k - 1} \cdot \frac{i}{T} \,\middle|\, i \in \{1, 2, \ldots, T\} \right\}.
\tag{9}
$$

To reduce quantization error, $T$ should be sufficiently large (e.g., 2048). However, computing the zero-point for a fixed

*Table 1.* Relative loss and runtime of NeUQI and NeUQI without coarse-to-fine search, compared with NeUQI without coarse-to-fine search and without transition-point reduction (baseline), including the baseline runtime, under 2-bit quantization on LLaMA-2 7B, averaged over all transformer blocks. Rel. L and Rel. T denote relative loss and relative runtime, respectively.

| Method | NeUQI(Ours) | | w/o coarse-to-fine | | baseline |
|---|---|---|---|---|---|
| | Rel. L | Rel. T | Rel. L | Rel. T | Time(s) |
| Q | 1.00193 | 0.027 | 1.00197 | 0.548 | 112 |
| K | 1.00271 | 0.026 | 1.00314 | 0.549 | 112 |
| V | 1.00006 | 0.026 | 1.00006 | 0.547 | 112 |
| O | 1.00001 | 0.026 | 1.00000 | 0.548 | 112 |
| Up | 1.00003 | 0.026 | 1.00002 | 0.53 | 308 |
| Gate | 1.00006 | 0.025 | 1.00004 | 0.531 | 308 |
| Down | 1.00000 | 0.028 | 1.00000 | 0.607 | 308 |

scale is expensive, making large $T$ incur substantial computational overhead. To address this issue, we adopt a coarse-to-fine search strategy. Specifically, we first perform a coarse search over $\mathcal{S}_{T_c}$ to obtain a coarse optimal scale $s^c$, which localizes the region of the optimal scale, where $T_c = O(\sqrt{T})$. We then conduct a finer-grained search over an approximately $T/T_c$-sized neighborhood of $s^c$ within $\mathcal{S}_T$ to obtain a refined scale estimate. As a result, the number of zero-point computations is reduced to $O(\sqrt{T})$.

We further compare relative loss and runtime across three variants: NeUQI, NeUQI without coarse-to-fine search, and NeUQI without both coarse-to-fine search and transition-point reduction, with the last serving as the baseline. Experiments are conducted under 2-bit quantization on LLaMA 2 7B and averaged over all transformer blocks, as shown in Table 1. Without any optimization, initializing a single query projection matrix takes 112 seconds. Transition-point reduction alone reduces runtime by nearly 50%, with an expected reduction of about 75% under 3-bit quantization. Further incorporating coarse-to-fine search yields an additional ∼95% runtime reduction relative to the variant without it. Overall, these optimizations introduce only a modest loss increase.

## 5. Experiment

### 5.1. Baselines and Evaluation

We experiment with NeUQI on three commonly-used LLM families, covering different sizes: LLaMA 2 (7B, 13B, 70B) (Touvron et al., 2023), LLaMA 3 (8B, 70B) (Grattafiori et al., 2024) and Qwen 2.5 (7B, 14B, 32B, 72B) (Yang et al., 2025). In the context of **uniform quantization**, we compare with several representative weight-only post-training methods that follow this scheme, including GPTQ (Frantar et al., 2023), MagR (Zhang et al., 2024), GP-TAQ (Li et al., 2025), and LeanQuant (Zhang & Shrivastava, 2025), together with three initialization strategy baselines for analytical purposes. We also evaluate NeUQI combined with the Hadamard transform, a representative instance of

*Table 2.* Perplexity on Wiki2 and C4, and zero-shot accuracy on five benchmarks (averaged as **Acc**) for 2-bit channel-wise quantized models. ↑/↓ indicate whether higher or lower is better, and **Size** denotes the model size. † marks results without extra coordinate descent iterations used in their original paper for the fair comparison.

| Model | Size | Bits | Method | Wiki2↓ | C4↓ | ArcC↑ | ArcE↑ | HellaS↑ | PiQA↑ | WinoG↑ | Acc↑ |
|---|---|---|---|---|---|---|---|---|---|---|---|
| LLaMA 2 | 7B | 2 | GPTQ | 6953 | 2592 | 21.93 | 25.63 | 25.89 | 52.23 | 49.72 | 35.08 |
| | | | GPTAQ | 1269 | 246 | 21.76 | 26.89 | 25.74 | 53.32 | 47.12 | 34.97 |
| | | | MagR† | 129 | 47.55 | 21.42 | 34.97 | 32.29 | 58.27 | 50.75 | 39.54 |
| | | | NeUQI | **17.14** | **17.50** | **23.98** | **51.73** | **36.04** | **65.89** | **58.56** | **47.24** |
| | 13B | 2 | GPTQ | 1735 | 433 | 21.76 | 26.52 | 25.85 | 52.56 | 51.30 | 35.60 |
| | | | GPTAQ | 145 | 62.05 | 20.56 | 26.89 | 26.99 | 53.43 | 51.85 | 35.95 |
| | | | MagR† | 47.98 | 27.94 | 23.21 | 26.68 | 36.70 | 52.99 | 54.14 | 38.74 |
| | | | NeUQI | **13.72** | **14.39** | **26.19** | **55.18** | **37.37** | **65.94** | **59.12** | **48.76** |
| | 70B | 2 | GPTQ | 60.29 | 46.11 | 19.97 | 28.07 | 27.05 | 53.75 | 50.04 | 35.78 |
| | | | GPTAQ | 40.09 | 28.37 | 21.59 | 28.91 | 28.98 | 54.08 | 51.46 | 37.00 |
| | | | MagR† | 68.62 | 13.95 | 39.59 | 71.21 | 52.81 | 76.33 | 69.06 | 61.80 |
| | | | NeUQI | **7.03** | **8.88** | **44.71** | **75.97** | **53.62** | **77.64** | **73.72** | **65.13** |
| LLaMA 3 | 8B | 2 | GPTQ | >1e4 | >1e4 | 20.90 | 23.91 | 26.04 | 53.48 | 48.86 | 34.64 |
| | | | GPTAQ | >1e4 | 4409 | 20.99 | 25.29 | 25.88 | 52.99 | 48.46 | 34.72 |
| | | | MagR | 387 | 140 | 18.43 | 30.09 | 27.60 | 55.98 | 50.12 | 36.45 |
| | | | NeUQI | **64.47** | **39.41** | **24.83** | **52.10** | **37.50** | **63.00** | **59.04** | **47.30** |
| | 70B | 2 | GPTQ | >1e4 | >1e4 | 20.73 | 25.76 | 25.76 | 52.45 | 48.46 | 34.63 |
| | | | GPTAQ | >1e4 | >1e4 | 22.70 | 24.87 | 25.78 | 53.21 | 51.22 | 35.56 |
| | | | MagR | >1e4 | >1e4 | 21.59 | 25.59 | 25.51 | 52.99 | 48.46 | 34.83 |
| | | | NeUQI | **56.21** | **42.06** | **25.68** | **53.24** | **36.98** | **61.04** | **55.49** | **46.49** |
| Qwen 2.5 | 7B | 2 | GPTQ | 2332 | 719 | 21.42 | 26.52 | 25.70 | 50.92 | 48.93 | 34.70 |
| | | | GPTAQ | 637 | 244 | 23.21 | 25.34 | 26.49 | 52.23 | 46.96 | 34.85 |
| | | | MagR | **25.38** | **24.83** | **23.12** | 42.09 | 37.76 | **60.99** | 51.30 | 43.05 |
| | | | NeUQI | 37.46 | 28.77 | 22.10 | **47.85** | **39.56** | **60.99** | **58.48** | **45.80** |
| | 14B | 2 | GPTQ | 3852 | 1056 | 23.12 | 25.42 | 25.90 | 50.92 | 51.93 | 35.46 |
| | | | GPTAQ | 1363 | 211 | 22.35 | 25.63 | 25.6 | 51.31 | 49.8 | 34.94 |
| | | | MagR | **18.36** | 19.36 | 23.21 | 44.74 | 38.06 | 63.44 | 52.88 | 44.47 |
| | | | NeUQI | 23.58 | **18.38** | **37.88** | **69.32** | **46.86** | **72.31** | **67.40** | **58.76** |
| | 32B | 2 | GPTQ | 308 | 141 | 21.42 | 26.09 | 25.11 | 51.20 | 49.41 | 34.65 |
| | | | GPTAQ | 277 | 86.61 | 20.82 | 23.86 | 27.08 | 52.23 | 51.22 | 35.04 |
| | | | MagR | **13.16** | **14.72** | 25.94 | 48.91 | 45.25 | 68.06 | 56.51 | 48.93 |
| | | | NeUQI | 18.17 | 15.77 | **41.55** | **71.63** | **53.38** | **76.01** | **72.77** | **63.07** |
| | 72B | 2 | GPTQ | 1127 | 281 | 22.95 | 25.00 | 25.46 | 51.31 | 48.70 | 34.68 |
| | | | GPTAQ | 576 | 105 | 21.59 | 23.40 | 27.87 | 51.74 | 49.09 | 34.74 |
| | | | MagR | 19.25 | 13.83 | 31.91 | 59.64 | 51.13 | 72.74 | 59.59 | 55.00 |
| | | | NeUQI | **10.79** | **11.36** | **48.04** | **78.11** | **56.18** | **78.56** | **75.14** | **67.21** |

the transformation-based techniques exemplified by QuIP, DuQuant, and FrameQuant, under both weight-only and weight-activation quantization. We evaluate NeUQI with a lightweight distillation setting and compare it with the distillation-based methods PV-tuning (Malinovskii et al., 2024) and OmniQuant (Shao et al., 2024). In this setting, all continuous parameters except the embedding and language model head are updated, and the objective is defined as the mean squared error between the final-layer hidden states of the full-precision and quantized models. In addition, under a fine-tuning regime, we evaluate EfficientQAT initialized with NeUQI and compare it with EfficientQAT (Chen et al., 2025). Following the previous work, all the quantized models are evaluated by measuring perplexity on the WikiText2 (**Wiki2**) (Merity et al., 2017)

and **C4** (Raffel et al., 2020) validation sets, and zero-shot accuracy on five benchmarks: ARC-easy (**ArcE**), ARC-challenge (**ArcC**) (Clark et al., 2018), **PiQA** (Bisk et al., 2020), HellaSwag (**HellaS**) (Zellers et al., 2019), and Wino-Grande (**WinoG**) (Sakaguchi et al., 2021).

### 5.2. Implementation Settings

We evaluate weight-only quantization using channel- and group-wise schemes at 2, 3, and 4 bits, with the group size fixed to 128 as in prior work. For weight–activation quantization, we use channel-wise weights at 2 and 4 bits, with activations quantized to 4 bits using token-wise dynamic Min-Max initialization. We denote a configuration as WxAy, where x and y are the bit widths of weights and activations,

*Table 3.* Results of LLaMA 2 7B with W2A16, W2A4, and W4A4 under the Hadamard transform.

| Setting | Method | Wiki2↓ | C4↓ | Acc↑ |
|---------|--------|--------|-----|------|
| **W2A16** | GPTQ | 759 | 277 | 34.95 |
| | GPTAQ | 64.88 | 39.36 | 37.14 |
| | MagR | 13.79 | 15.05 | 48.01 |
| | NeUQI | **12.41** | **13.22** | **52.91** |
| **W2A4** | GPTQ | 1098 | 463 | 34.88 |
| | GPTAQ | 85.65 | 50.17 | 36.78 |
| | MagR | 20.42 | 20.54 | 42.95 |
| | NeUQI | **13.63** | **14.91** | **50.33** |
| **W4A4** | GPTQ | 6.14 | 7.75 | 61.67 |
| | GPTAQ | 6.20 | 7.73 | 61.93 |
| | MagR | 6.02 | 7.59 | 62.69 |
| | NeUQI | **5.99** | **7.57** | **62.77** |

respectively. Across all experiments, we employ GPTQ with column quantization ordered by decreasing activation L2-norm as the weight transformation method, ensuring fair comparison across different approaches. We follow GPTQ for calibration, using 128 C4 samples with 2048 tokens for LLaMA 2 and 4096 tokens for LLaMA 3 and Qwen 2.5. We perform distillation with 256 C4 samples for one epoch. We set NeUQI candidate space hyperparameters to $T = 2048$ and $T_c = 64$. For fairness, MagR is evaluated without extra coordinate descent iterations used in the original paper. More implementation details are included in Appendix D.

## 5.3. Main Results

Under the most challenging 2-bit channel-wise quantization setting, our NeUQI achieves significant improvements over GPTQ, GPTAQ and MagR, as shown in Table 2. Although the recent MagR method delivers acceptable performance on the Qwen 2.5 family, especially on perplexity, it performs poorly on the LLaMA family. In contrast, NeUQI consistently demonstrates strong performance across different model architectures and sizes. Similar conclusions hold under the 2-bit group-wise quantization setting, with detailed results shown in Table 12 in Appendix H.

Moreover, as the bit-width increases, the performance gap between different methods gradually narrows, as shown in Table 13, Table 14, Table 15 and Table 16 in Appendix H. Our NeUQI continues to show stable advantages under the 3-bit setting, maintaining robustness across architectures. As for 4-bit, where all methods closely match the original non-quantized model, further improvements are limited. Nevertheless, NeUQI remains competitive and exhibits slight advantages in certain cases. In addition, with our optimized implementation, NeUQI incurs acceptable runtime overhead, as detailed in Appendix F.

**Hadamard Transform**  We further evaluate NeUQI in combination with the representative transformation-based

*Table 4.* Results of different initialization methods on LLaMA 2 and 3 family with 2-bit channel-wise quantization. † marks results from the original paper.

| Model | Method | Wiki2↓ | C4↓ | Acc↑ |
|-------|--------|--------|-----|------|
| LLaMA 2 7B | Min-Max+ | 498 | 136 | 36.20 |
| | MSE | 65.42 | 93.58 | 44.19 |
| | LeanQuant† | 25.69 | 27.11 | 42.43 |
| | Int-Search | 26.26 | 24.15 | 43.85 |
| | NeUQI | **17.14** | **17.50** | **47.24** |
| LLaMA 2 13B | Min-Max+ | 32.19 | 25.96 | 39.17 |
| | MSE | 155 | 223 | 36.42 |
| | LeanQuant† | 24.43 | 20.92 | 47.46 |
| | Int-Search | 15.67 | 16.69 | 47.60 |
| | NeUQI | **13.72** | **14.39** | **48.76** |
| LLaMA 2 70B | Min-Max+ | 13.33 | 14.08 | 47.00 |
| | MSE | 11.98 | 13.16 | 58.10 |
| | LeanQuant† | 7.92 | 10.84 | – |
| | Int-Search | 8.71 | 10.49 | 61.61 |
| | NeUQI | **7.03** | **8.88** | **65.13** |
| LLaMA 3 8B | Min-Max+ | 3016 | 716 | 35.18 |
| | MSE | 147 | 57.94 | 37.82 |
| | LeanQuant† | – | – | 39.18 |
| | Int-Search | 93.64 | 47.76 | 40.78 |
| | NeUQI | **64.47** | **39.41** | **47.30** |

technique, the Hadamard transform, under the weight-only setting W2A16 and the weight–activation settings W2A4 and W4A4. These experiments demonstrate that NeUQI can be effectively integrated with transformation-based techniques to further enhance performance. As shown in Table 3, NeUQI consistently outperforms baselines in these settings.

## 5.4. Bit Analysis

**Integer Constraint**  We conduct an ablation study on initialization and the integer zero-point constraint, and compare our approach with several improved initialization methods that preserve the integer zero-point constraint, including Min-Max+[1] and MSE[2], as well as search-based LeanQuant and Int-Search, the latter sharing NeUQI's loss function 5. The differences between LeanQuant and Int-Search are discussed in Appendix D.3. As shown in Table 4, these methods improve upon Table 2 but still fall short of NeUQI. In the 2-bit setting, NeUQI gains superior results with less than 0.01 increase in average bit-width. This suggests that the common integer zero-point assumption is overly restrictive for optimization-based quantization and deserves further study.

**Bit Fairness**  For channel-wise quantization, removing the integer constraint leads to only a negligible increase

---

[1]A variant derived from correcting the flawed intuition of Min-Max, as detailed in Appendix A.

[2]A legacy MSE-based grid search method implemented in the GPTQ repository.

*Table 5.* Performance comparison on LLaMA 2 7B under 2-bit group-wise quantization with approximately equal average bit-width.

| Group | Avg Bits | Method | Wiki2↓ | C4↓ | Acc↑ |
|---|---|---|---|---|---|
| 64 | 2.28 | GPTQ | 16.03 | 16.04 | 46.14 |
| 64 | 2.28 | GPTAQ | 11.88 | 12.56 | 49.73 |
| 64 | 2.28 | MagR | 14.58 | 15.44 | 48.30 |
| 64 | 2.28 | Int-Search | 14.67 | 15.96 | 49.38 |
| 128 | 2.25 | NeUQI | 12.35 | 13.48 | 51.33 |
| 64 | 2.5 | NeUQI | 10.61 | 12.03 | 53.22 |

*Table 6.* Results of distillation on LLaMA 2 7B with 2-bit quantization and group size 128. † marks results from Li & Panda (2024), ‡ marks group size 64, § marks lightweight distillation, and **Tokens** indicates training set size times epochs.

| Method | Tokens | Wiki2↓ | C4↓ | Acc↑ |
|---|---|---|---|---|
| GPTQ[§] | $\sim 0.5M \times 1$ | 14.95 | 13.31 | 48.05 |
| OmniQuant[†] | $\sim 0.25M \times 20$ | 11.06 | 16.34 | 47.59 |
| OmniQuant[†‡] | $\sim 0.25M \times 20$ | 9.62 | 13.79 | - |
| PV-tuning[†] | $\sim 1B \times 1$ | 8.49 | 10.78 | 52.17 |
| NeUQI[§] | $\sim 0.5M \times 1$ | **8.38** | **9.81** | **56.77** |

in average bit-width. In 2-bit quantization with group size 128, the average bit-width is about 2.14 with the constraint and 2.25 without it. To ensure fair comparison, we also evaluate methods with the integer constraint at group size 64, which yields an average bit-width of about 2.28. As shown in Table 5, NeUQI delivers superior performance even at a lower average bit-width. Furthermore, double quantization (Dettmers et al., 2023) can further reduce the memory footprint of quantization parameters, mitigating the overhead of removing the integer constraint.

## 5.5. Discussions

**Distillation and Fine-tuning**  As shown in Table 6, applying our lightweight distillation to GPTQ yields performance comparable to OmniQuant, while applying it to NeUQI surpasses both PV-tuning and OmniQuant. Notably, this is achieved with only about 0.5M tokens, whereas PV-tuning and OmniQuant require more memory, incur longer per-step runtimes than standard fine-tuning, and demand substantially more data. These results highlight the importance of better initialization, as poor initialization appears difficult to remedy even with heavy fine-tuning. Moreover, methods that go beyond the first constraint can benefit from strong non-fine-tuning PTQ approaches such as NeUQI.

To further demonstrate the advantage of the better initialization within fine-tuning methods, we compare the very strong fine-tuning method EfficientQAT with a modified version that replaces only the quantization parameter initialization with NeUQI. The results shown in Table 7 indicate that even for such a strong fine-tuning method, NeUQI still provides

*Table 7.* Results of EfficientQAT and its NeUQI-initialized variant under 2-bit quantization with group size 128 on LLaMA 2 7B and LLaMA 3 8B. † marks results from the original paper. § denotes freezing the scale in the first stage of EfficientQAT.

| Model | Method | Wiki2↓ | C4↓ | Acc↑ |
|---|---|---|---|---|
| LLaMA 2 7B | EfficientQAT† | 7.19 | 8.79 | 59.50 |
| | +Floating $z$ | 7.02 | 8.72 | 59.37 |
| | +NeUQI | 6.81 | 8.41 | 60.01 |
| | +NeUQI[§] | **6.73** | **8.36** | **60.11** |
| LLaMA 3 8B | EfficientQAT† | 9.80 | 13.22 | 59.37 |
| | +Floating $z$ | 9.72 | 13.22 | 59.22 |
| | +NeUQI | 9.14 | 12.51 | 60.68 |
| | +NeUQI[§] | **8.87** | **12.32** | **61.49** |

*Table 8.* Performance comparison between quantized and non-quantized models based on the Qwen 2.5 family with comparable main body memory usages.

| Method | Memory | Size | Bits | Wiki2↓ | C4↓ | Acc↑ |
|---|---|---|---|---|---|---|
| - | 2.62 GiB | 1.5B | 16 | **8.58** | 12.54 | 61.30 |
| GPTQ | 2.45 GiB | 7B | 3 | 10.73 | 13.90 | 58.95 |
| NeUQI | 2.45 GiB | 7B | 3 | 8.64 | **11.72** | **65.13** |
| - | 5.55 GiB | 3B | 16 | 7.44 | 11.15 | 64.99 |
| GPTQ | 4.96 GiB | 14B | 3 | 8.48 | 11.61 | 60.26 |
| NeUQI | 4.96 GiB | 14B | 3 | **6.82** | **9.89** | **69.12** |
| - | 13.05 GiB | 7B | 16 | 6.39 | 10.02 | 68.04 |
| GPTQ | 11.72 GiB | 32B | 3 | 7.21 | 10.36 | 64.38 |
| NeUQI | 11.72 GiB | 32B | 3 | **5.85** | **9.31** | **69.95** |
| - | 26.42 GiB | 14B | 16 | **4.93** | 8.75 | 71.46 |
| GPTQ | 26.36 GiB | 72B | 3 | 6.89 | 9.96 | 67.30 |
| NeUQI | 26.36 GiB | 72B | 3 | 4.99 | **8.37** | **73.07** |

further performance improvements. Moreover, with the high-quality initialization provided by NeUQI, freezing the scale in the first stage of EfficientQAT results in improved performance. This may be because the well-initialized scale is already close to a local optimum in this setting, making further scale updates unnecessary or even detrimental.

**Comparison with Non-Quantized Models**  With comparable memory usages, the quantized models using NeUQI consistently outperform the non-quantized ones, as shown in Table 8. Although experiments on larger models are constrained by computational resources, these results demonstrate the potential of NeUQI-based low-bit-width quantization when applied to ultra-large models, which often lack smaller variants and remain difficult to deploy in resource-limited environments even at 4-bit precision.

## 6. Conclusion

To the best of our knowledge, we are the first to identify the constraints of the *Min–Max formula*. By going beyond these constraints, NeUQI consistently outperforms existing methods and, with a reduced memory footprint, can even

surpass the integer-constrained upper bound and the corresponding non-quantized model. As an initialization method, NeUQI is also compatible with transformation-based techniques and fine-tuning, with experiments on distillation and fine-tuning further confirming its effectiveness. For future work, we will explore fine-tuning strategies beyond the first constraint to further enhance post-initialization performance under uniform quantization.

## Impact Statement

This paper presents work whose goal is to advance the field of machine learning. There are many potential societal consequences of our work, none of which we feel must be specifically highlighted here.

## Acknowledgements

This work was supported by Beijing Natural Science Foundation (L253001), Key Laboratory of Science, Technology and Standard in Press Industry (Key Laboratory of Intelligent Press Media Technology) and National Engineering Research Center of New Electronic Publishing Technologies. We appreciate the anonymous reviewers for their helpful comments. Xiaojun Wan is the contact author.

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

## A. Min-Max Intuitive Fallacy

Intuitively, Min-Max maps the entire observed range $[x_{\min}, x_{\max}]$ onto the low-error interval $[-zs, (-z + 2^k - 1)s]$. However, the more accurate and appropriate interval should be $[(-z - \frac{1}{2})s, (-z + 2^k - \frac{1}{2})s]$. Based on this correction, we simply propose an improved initialization strategy, **Min-Max+**, defining the quantization parameters as follows:

$$s = \frac{\max(x) - \min(x)}{2^k}, \quad z = -\left\lfloor \frac{\min(x)}{s} + \frac{1}{2} \right\rceil, \tag{10}$$

which matches the configuration that minimizes expected error when weights are ideally drawn independently and uniformly from an interval, as detailed with the theoretical analysis in Appendix B. Empirically, our simple adjustment leads to measurable gains, which confirms the flawed intuition of original Min-Max method. In real-world settings such as GPTQ quantization of the LLaMA 2 7B, Min-Max+ consistently outperforms Min-Max, as shown in Table 9.

*Table 9.* Comparison between Min–Max and Min–Max+ using GPTQ on the LLaMA-2-7B model. **Wx** denotes x-bit quantization and **Gy** denotes group size y.

| Setting | W3G128 | | W3 | | W2G128 | | W2 | |
|---|---|---|---|---|---|---|---|---|
| Method | Wiki2↓ | C4↓ | Wiki2↓ | C4↓ | Wiki2↓ | C4↓ | Wiki2↓ | C4↓ |
| Min-Max | 6.34 | 7.86 | 8.45 | 9.87 | 26.31 | 58.33 | 6953 | 2592 |
| Min-Max+ | **6.32** | **7.72** | **7.28** | **8.84** | **13.64** | **13.64** | **497** | **135** |

## B. Optimal Quantization Parameter

**Lemma B.1.** *If weights are independently and uniformly drawn from an interval $[a, b]$, then the optimal parameters to for $k$-bit uniform quantization to minimize expected quantization error $\mathbb{E}_{x \sim \mathcal{U}(a,b)}[(Q_{s,z}(x) - x)^2]$ are given by*

$$s = \frac{b - a}{2^k}, \quad z = -\left(\frac{a}{s} + \frac{1}{2}\right). \tag{11}$$

*Proof.* We relax the uniform spacing constraint on $\mathcal{Q}$ to allow non-uniform intervals, but will show the minimum MSE occurs when they are all equal. The expected quantization error is

$$\mathcal{E} = \frac{1}{b - a} \sum_{i=0}^{N-1} \int_{b_i}^{b_{i+1}} (x - q_i)^2 \, dx, \tag{12}$$

where $N = 2^k$, $a = b_0 < \cdots < b_N = b$, and each $[b_i, b_{i+1})$ is mapped to $q_i$.

Let

$$\Delta_i = b_{i+1} - b_i. \tag{13}$$

For fixed $\Delta_i$, minimizing the integral

$$\int_{b_i}^{b_{i+1}} (x - q_i)^2 \, dx \tag{14}$$

yields

$$q_i^* = \frac{1}{\Delta_i} \int_{b_i}^{b_{i+1}} x \, dx = \frac{b_i + b_{i+1}}{2}, \tag{15}$$

and thus

$$\int_{b_i}^{b_{i+1}} (x - q_i^*)^2 \, dx = \frac{\Delta_i^3}{12}. \tag{16}$$

Substituting equation 16 into equation 12 gives

$$\mathcal{E} = \frac{1}{12(b - a)} \sum_{i=0}^{N-1} \Delta_i^3. \tag{17}$$

Since $f(\Delta) = \Delta^3$ is strictly convex on $(0, \infty)$, Jensen's inequality shows $\mathcal{E}$ is minimized when all $\Delta_i$ are equal:

$$\Delta_i = \frac{b-a}{N} = \frac{b-a}{2^k}. \tag{18}$$

Therefore the optimal uniform quantizer parameters are

$$s = \Delta_i = \frac{b-a}{2^k}, \quad z = -\frac{q_0^*}{s} = -\left(\frac{a}{s} + \frac{1}{2}\right). \tag{19}$$

$\square$

## C. Zero-point Algorithms

Detailed zero-point algorithms corresponding to Section 4.2.1 are provided below.

---

**Algorithm 2** Optimal Zero-point over Eq 7

---

**Input** Samples $\{x_i\}_{i=1}^n$, sample-wise weights $\{h_i\}_{i=1}^n$, bit-width $k$
Initialize list of transition points: $\mathcal{T} \leftarrow [\,]$
**for** each sample $x_i$ **do**
    **for** $j = 0$ **to** $2^k - 2$ **do**
        $t \leftarrow j + \frac{1}{2} - x_i$
        $\delta_{i,j}(z) \leftarrow h_i\big(x_i + z - (j+1)\big)^2 - h_i\big(x_i + z - j\big)^2$
        Add $(t, \delta_{i,j}(z))$ to $\mathcal{T}$
    **end for**
**end for**
Sort $\mathcal{T}$ by transition point $t$
$\mathcal{L}^{\mathrm{I}}(z) \leftarrow \sum_i h_i(x_i + z)^2$
$(z^*, \mathcal{L}^*) \leftarrow (-\infty, +\infty)$
**for** each $(t, \delta(z))$ in $\mathcal{T}$ **do**
    $\mathcal{L}^{\mathrm{I}}(z) \leftarrow \mathcal{L}^{\mathrm{I}}(z) + \delta(z)$
    Let next transition point be $t'$ (or $+\infty$ if none)
    $z' \leftarrow \arg\min_{z \in [t,t']} \mathcal{L}^{\mathrm{I}}(z)$
    update $(z^*, \mathcal{L}^*)$ with $(z', \mathcal{L}^{\mathrm{I}}(z'))$ if $\mathcal{L}^{\mathrm{I}}(z') < \mathcal{L}^*$
**end for**
**Return** $z^*$

---

**Algorithm 3** Optimal Zero-point over Eq 8

---

**Input** Samples $\{x_i\}_{i=1}^n$, sample-wise weights $\{h_i\}_{i=1}^n$, bit-width $k$
Initialize transition list $\mathcal{T} \leftarrow [\,]$
**for** each sample $x_i$ **do**
    $t_{\mathrm{enter}} \leftarrow -\frac{1}{2} - x_i$
    $\delta_{\mathrm{enter}}(z) \leftarrow h_i(x_i + z)^2 - h_i \cdot \frac{1}{4}$
    $t_{\mathrm{exit}} \leftarrow 2^k - \frac{1}{2} - x_i$
    $\delta_{\mathrm{exit}}(z) \leftarrow h_i \cdot \frac{1}{4} - h_i(x_i + z - (2^k - 1))^2$
    Add $(t_{\mathrm{enter}}, \delta_{\mathrm{enter}}(z))$ and $(t_{\mathrm{exit}}, \delta_{\mathrm{exit}}(z))$ to $\mathcal{T}$
**end for**
Sort $\mathcal{T}$ by transition point $t$
$\mathcal{L}^{\mathrm{I}}(z) \leftarrow \sum_i h_i(x_i + z)^2$
$(z^S, \mathcal{L}^S) \leftarrow (-\infty, +\infty)$
**for** each $(t, \delta(z))$ in $\mathcal{T}$ **do**
    $\mathcal{L}^{\mathrm{I}}(z) \leftarrow \mathcal{L}^{\mathrm{I}}(z) + \delta(z)$
    Let next transition point be $t'$ (or $+\infty$ if none)
    $z' \leftarrow \arg\min_{z \in [t,t']} \mathcal{L}^{\mathrm{I}}(z)$
    update $(z^S, \mathcal{L}^S)$ with $(z', \mathcal{L}^{\mathrm{I}}(z'))$ if $\mathcal{L}^{\mathrm{I}}(z') < \mathcal{L}^S$
**end for**
**Return** $z^S$

---

---

**Algorithm 4** Optimal Zero-point over Eq. 7 in Limited Interval

---

**Input** Samples $\{x_i\}_{i=1}^n$, sample-wise weights $\{h_i\}_{i=1}^n$, bit-width $k$, interval $[z^S - 1, z^S + 1]$
Initialize transition list $\mathcal{T} \leftarrow [\,]$
**for** each sample $x_i$ **do**
    $j_1 = \lceil z^S - 1 + x_i - \frac{1}{2} \rceil$
    $t_1 = j_1 + \frac{1}{2} - x_i$ {First possible transition point in $[z^S - 1, z^S + 1)$}
    $\delta_1(z) \leftarrow h_i(x_i + z - (j_1 + 1))^2 - h_i(x_i + z - j_1)^2$
    **if** $0 \le j_1 < 2^k - 1$ **then**
        Add $(t_1, \delta_1(z))$ to $\mathcal{T}$
    **end if**
    $j_2 = j_1 + 1$
    $t_2 = j_2 + \frac{1}{2} - x_i$ {Second possible transition point in $[z^S - 1, z^S + 1)$}
    $\delta_2(z) \leftarrow h_i(x_i + z - (j_2 + 1))^2 - h_i(x_i + z - j_2)^2$
    **if** $0 \le j_2 < 2^k - 1$ **then**
        Add $(t_2, \delta_2(z))$ to $\mathcal{T}$
    **end if**
    {Since $(\lceil z^S - 1 + x_i - \frac{1}{2} \rceil + 2) + \frac{1}{2} - x_i \ge z^S + 1$, there are at most two transition points in the interval $[z^S - 1, , z^S + 1)$.}
**end for**
Sort $\mathcal{T}$ by transition point $t$
$\mathcal{L}^{\mathrm{I}}(z) \leftarrow \sum_i h_i \Big( x_i + z - \mathrm{clip}(\lfloor x_i + z^S - 1 \rceil, 0, 2^k - 1) \Big)^2$
$(t_{\text{first}},\ \delta_{\text{first}}) \leftarrow$ first element of $\mathcal{T}$
$z' \leftarrow \arg\min_{z \in [z^S - 1,\, t_{\text{first}}]} \mathcal{L}^{\mathrm{I}}(z)$
$(z^*, \mathcal{L}^*) \leftarrow (z', \mathcal{L}^{\mathrm{I}}(z'))$
**for** each $(t, \delta(z))$ in $\mathcal{T}$ **do**
    $\mathcal{L}^{\mathrm{I}}(z) \leftarrow \mathcal{L}^{\mathrm{I}}(z) + \delta(z)$
    Let next transition point be $t'$ (or $z^S + 1$ if none) {Consider the value at $z^S + 1$}
    $z' \leftarrow \arg\min_{z \in [t, t']} \mathcal{L}^{\mathrm{I}}(z)$
    update $(z^*, \mathcal{L}^*)$ with $(z', \mathcal{L}^{\mathrm{I}}(z'))$ if $\mathcal{L}^{\mathrm{I}}(z') < \mathcal{L}^*$
**end for**
**Return** $z^*$

---

# D. Implementation Details

## D.1. Setting

### D.1.1. CALIBRATION STAGE

During the calibration stage, we follow the implementation of the official GPTQ repository,[3] and draw the same 128 samples from the C4 dataset for quantization calibration. For implementation convenience, we uniformly adopt `bfloat16` for all experiments. The token length of each sample is determined based on the characteristics and design intent of the target family. Especially for the LLaMA 2 family, each sample consists of 2048 tokens, consistent with the configurations adopted in GPTQ and LeanQuant. For the Qwen 2.5 models, each sample comprises 4096 tokens, as indicated in the Qwen 2.5 Technical Report, which specifies a 4096-token context length in its base configuration before context expansion. The same token length setting is applied to the LLaMA 3 family to ensure consistency. All experiments are conducted on NVIDIA A40 GPUs and NVIDIA L40 GPUs.

### D.1.2. DISTILLATION STAGE

In contrast to the calibration stage, the distillation utilizes 256 samples from the C4 dataset. The token lengths per sample remain consistent with those used during calibration. Furthermore, the computation precision remains `bfloat16`, identical to that in the calibration phase.

For the broader fine-tuning setup, we employ the AdamW optimizer with zero weight decay and momentum parameters set as $\beta = (0.9, 0.95)$. The learning rate is chosen via grid search over five candidate values: 1e-5, 3e-5, 1e-4, 3e-4, and 1e-3. We apply a cosine learning rate scheduler with a 10% warm-up ratio. In terms of hardware and infrastructure, we fine-tune LLaMA 2 7B in mixed precision using DeepSpeed[4], with batch size of 1 on a single NVIDIA A40 GPU.

### D.1.3. EVALUATION

Regarding perplexity evaluation, we follow the procedure implemented in the official GPTQ repository. Perplexity scores are computed on the validation sets of the C4 and WikiText2 datasets, using the dataset identifiers c4 and wikitext2 as specified in the original implementation.

For zero-shot accuracy evaluation, we adopt the lm-evaluation-harness (Gao et al., 2024), a standardized and extensible framework for evaluating language models across diverse tasks. Inference is conducted using vLLM (Kwon et al., 2023) as the backend to ensure high throughput and memory efficiency. We evaluate five benchmark tasks: ARC-Easy, ARC-Challenge, WinoGrande, HellaSwag, and PIQA, whose corresponding identifiers in lm-evaluation-harness are arc_easy, arc_challenge, winogrande, hellaswag, and piqa, respectively.

## D.2. Quantization

We adopt the quantization strategy implemented in the official GPTQ repository. Specifically, we perform sequential quantization by quantizing layers in the order of the model forward pass. Each layer receives inputs from the already quantized preceding layers rather than from the original model, ensuring that the quantization is based on realistic activation distributions that more closely resemble those encountered during inference. Furthermore, we conduct the entire quantization process under full-precision settings to ensure numerical stability. For NeUQI, the grid search over the scale hyperparameter is conducted with $T = 2048$ and $T_c = 64$.

## D.3. Difference between LeanQuant and Int-Search

LeanQuant$_{aff}$, short for LeanQuant, and Int-Search adopt the same loss formulation of the form

$$\mathcal{L}(s, z) = \sum_i h_i (Q_{s,z}(w_i) - w_i)^2, \tag{20}$$

but differ fundamentally in how the weight importance $h_i$ is defined and how to find the optimal scale and zero-point.

---

[3] https://github.com/IST-DASLab/gptq
[4] https://github.com/microsoft/DeepSpeed

**Weight importance**  In LeanQuant, the weight importance $h_i$ is inspired by the objective used in the first step of GPTQ, which selects the optimal quantization point by minimizing

$$\frac{(Q_{s,z}(w_i) - w_i)^2}{(\boldsymbol{H}^{-1})_{ii}},\tag{21}$$

leading to $h_i = ((\boldsymbol{H}^{-1})_{ii})^{-1}$, and further generalized to $h_i = ((\boldsymbol{H}^{-1})_{ii})^{-p}$ with a tunable strength parameter $p$. In contrast, Int-Search adopts the diagonal loss approximation by directly setting $h_i = H_{ii}$, avoiding the need to invert the Hessian matrix.

**Parameter Search**  LeanQuant still operates within the *Min-Max formula*: its grid search essentially enumerates different min and max values, from which they derive the scale and zero-point using the standard *Min-Max formula*, with a time complexity of $\mathcal{O}(T^2 n)$, where $T$ (typically 2048) denotes the number of grid points and $n$ the number of weights. In contrast, Int-Search employs a grid search over candidate scale values, with the zero-point constrained to be a $k$-bit unsigned integer. A simple implementation of this scale-based search has a time complexity of $\mathcal{O}(T2^k n)$. To further accelerate initialization, we identify a more efficient alternative by switching the search perspective from scale to zero-point. For a fixed zero-point, the objective remains a piecewise quadratic function of the scale, similar to the case discussed in Section 4.2, which enables an optimized search using a similar method with total complexity reduced to $\mathcal{O}(2^k \cdot n2^k \log(n2^k))$, which is independent of $T$.

## E. Hardware Support for Floating-Point Zero-Points

For hardware support, the BitBLAS library[5], which has been integrated into the widely used inference framework vLLM[6], provides compatibility with mainstream GPUs such as A100 and RTX 4090, and supports zero-points represented in floating-point format when the parameter `zeros_mode` is set to `original`, as specified in the BitBLAS Python API[7]. This indicates that our proposed NeUQI has been supported by GPU kernels implemented in BitBLAS, without requiring specialized hardware modifications. Additionally, similar support is available in some other existing GPU quantization libraries.

## F. Quantization Process Runtime

Table 10 summarizes the quantization process runtime of baseline methods and NeUQI applied to the LLaMA-2 family under 2-bit channel-wise quantization on a single NVIDIA A40 GPU. Through our algorithmic design, NeUQI maintains a well-balanced quantization runtime while achieving considerable performance improvements, which facilitates further research and exploration. Moreover, the quantization process directly produces **a fully quantized model ready for immediate use**, without incurring any additional inference-time overhead.

*Table 10.* Quantization process runtime comparison of different quantization methods on the LLaMA-2 family under 2-bit channel-wise quantization using a single NVIDIA A40 GPU.

| Method | 7B (minutes) | 13B (minutes) | 70B (minutes) |
|--------|------------|-------------|-------------|
| GPTQ   | 8.75       | 15.70       | 76.72       |
| GPTAQ  | 14.35      | 27.23       | 178.08      |
| MagR   | 21.87      | 49.70       | 396.45      |
| NeUQI  | 29.75      | 53.03       | 292.93      |

## G. Another Transformation-based Method

We further evaluate another transformation-based quantization method, QuIP (Chee et al., 2023), and compare it with its NeUQI-enhanced variant. As shown in Table 11, incorporating NeUQI consistently improves the performance of QuIP. This observation suggests that, even for transformation-based methods that explicitly smooth the weight distribution before quantization, the initialization of quantization parameters remains a critical factor affecting the final quantization quality.

---

[5]https://github.com/microsoft/BitBLAS
[6]https://github.com/vllm-project/vllm
[7]https://github.com/microsoft/BitBLAS/blob/main/docs/PythonAPI.md

*Table 11.* Comparison between QuIP and its NeUQI-enhanced variant under 2-bit quantization. Results are reported for LLaMA 2 7B, LLaMA 2 13B, and LLaMA 3 8B.

| Model | Bits | Method | Wiki2↓ | C4↓ | ArcC↑ | ArcE↑ | HellaS↑ | PiQA↑ | WinoG↑ | Acc↑ |
|---|---|---|---|---|---|---|---|---|---|---|
| LLaMA 2 7B | 2 | QuIP | 149.30 | 110.38 | 20.05 | 28.24 | 27.10 | 54.95 | 50.59 | 36.19 |
| | | +NeUQI | **15.24** | **16.98** | **26.37** | **56.99** | **41.09** | **68.06** | **57.70** | **50.04** |
| LLaMA 2 13B | 2 | QuIP | 13.18 | 14.31 | 27.05 | 56.90 | 40.06 | 66.70 | 56.35 | 49.41 |
| | | +NeUQI | **8.63** | **10.34** | **32.59** | **66.79** | **47.60** | **73.72** | **63.93** | **56.93** |
| LLaMA 3 8B | 2 | QuIP | 150.27 | 121.28 | 20.82 | 26.77 | 27.99 | 53.26 | 48.93 | 35.56 |
| | | +NeUQI | **33.07** | **33.33** | **19.97** | **35.31** | **35.94** | **59.36** | **58.80** | **41.87** |

## H. Full Results

This appendix provides full quantization results. Table 12 presents detailed results of 2-bit quantization with a group size of 128 for LLaMA and Qwen families. Tables 13, 14, 15 and 16 report detailed results for 3-bit and 4-bit quantization of LLaMA 2, LLaMA 3, and Qwen 2.5 families, respectively.

*Table 12.* Results of 2-bit group-wise quantization with group size of 128 on the LLaMA 2, LLaMA 3, and Qwen 2.5 families. † denotes that the calibration set used differs from the original paper.

| Model | Size | Bits | Method | Wiki2↓ | C4↓ | ArcC↑ | ArcE↑ | HellaS↑ | PiQA↑ | WinoG↑ | Acc↑ |
|---|---|---|---|---|---|---|---|---|---|---|---|
| LLaMA 2 | 7B | 2 | GPTQ | 26.31 | 23.52 | 22.70 | 36.91 | 34.39 | 60.50 | 54.62 | 41.82 |
| | | | GPTAQ | 15.80 | 15.11 | 25.85 | 48.44 | 38.45 | 66.97 | 58.41 | 47.63 |
| | | | MagR† | 15.41 | 15.59 | 24.66 | 47.90 | 39.19 | 65.89 | 58.80 | 47.29 |
| | | | NeUQI | **12.35** | **13.48** | **25.26** | **59.72** | **40.71** | **69.64** | **61.33** | **51.33** |
| | 13B | 2 | GPTQ | 12.50 | 13.16 | 26.54 | 53.58 | 42.00 | 68.34 | 55.01 | 49.09 |
| | | | GPTAQ | 10.35 | 11.22 | 29.95 | 58.25 | 45.24 | 70.51 | 62.19 | 53.23 |
| | | | MagR† | 17.15 | 11.35 | 30.46 | 61.74 | 45.72 | 71.60 | 60.62 | 54.03 |
| | | | NeUQI | **8.82** | **10.60** | **35.32** | **67.72** | **45.98** | **73.12** | **68.59** | **58.15** |
| | 70B | 2 | GPTQ | 7.04 | 8.62 | 38.48 | 70.20 | 53.00 | 75.14 | 69.93 | 61.35 |
| | | | GPTAQ | 6.77 | 8.27 | 36.69 | 67.85 | 52.13 | 74.59 | 70.96 | 60.44 |
| | | | MagR† | 10.87 | 9.21 | 41.89 | 68.64 | 55.13 | 68.50 | 73.64 | 61.56 |
| | | | NeUQI | **5.60** | **7.51** | **47.53** | **77.86** | **56.90** | **78.24** | **75.53** | **67.21** |
| LLaMA 3 | 8B | 2 | GPTQ | 210 | 121 | 19.54 | 28.91 | 26.94 | 54.73 | 51.62 | 36.35 |
| | | | GPTAQ | 81.25 | 51.94 | 18.94 | 34.93 | 31.19 | 59.14 | 50.28 | 38.90 |
| | | | MagR | 61.71 | 74.78 | 18.43 | 33.80 | 28.72 | 56.69 | 51.70 | 37.87 |
| | | | NeUQI | **28.90** | **26.83** | **23.29** | **50.04** | **36.04** | **63.38** | **59.91** | **46.53** |
| | 70B | 2 | GPTQ | 26.25 | 28.59 | 22.27 | 38.80 | 34.61 | 61.04 | 55.41 | 42.43 |
| | | | GPTAQ | 19.67 | 20.20 | 20.39 | 38.13 | 40.77 | 60.45 | 59.75 | 43.90 |
| | | | MagR | 30.86 | 55.42 | 26.71 | 51.56 | 40.78 | 67.74 | 64.56 | 50.27 |
| | | | NeUQI | **10.75** | **14.40** | **39.68** | **71.30** | **48.66** | **74.05** | **70.56** | **60.85** |
| Qwen 2.5 | 7B | 2 | GPTQ | 21.66 | 22.77 | 23.72 | 43.43 | 39.32 | 62.02 | 53.75 | 44.45 |
| | | | GPTAQ | 19.06 | 20.96 | 27.3 | 46.63 | 41.69 | 66.92 | 56.59 | 47.83 |
| | | | MagR | **13.65** | **16.56** | 30.46 | 61.95 | 43.88 | 68.50 | 60.30 | 53.02 |
| | | | NeUQI | 14.43 | 16.65 | **36.26** | **67.76** | **45.56** | **71.55** | **65.04** | **57.23** |
| | 14B | 2 | GPTQ | 18.62 | 19.22 | 22.95 | 42.80 | 38.38 | 61.97 | 51.62 | 43.54 |
| | | | GPTAQ | 15.89 | 16.7 | 25.6 | 51.98 | 42.98 | 67.79 | 58.09 | 49.29 |
| | | | MagR | 12.94 | 14.09 | 28.33 | 56.57 | 44.52 | 70.40 | 60.77 | 52.12 |
| | | | NeUQI | **10.57** | **13.61** | **43.52** | **75.63** | **50.26** | **74.92** | **72.30** | **63.32** |
| | 32B | 2 | GPTQ | 10.95 | 13.36 | 30.38 | 59.30 | 47.56 | 71.38 | 56.59 | 53.04 |
| | | | GPTAQ | 11.15 | 13.2 | 32.51 | 64.14 | 48.71 | 74.32 | 59.12 | 55.76 |
| | | | MagR | 8.82 | 11.65 | 36.69 | 69.57 | 52.23 | 75.57 | 64.40 | 59.69 |
| | | | NeUQI | **8.51** | **11.41** | **46.08** | **76.39** | **55.86** | **78.02** | **75.14** | **66.30** |
| | 72B | 2 | GPTQ | 10.35 | 12.27 | 35.49 | 63.51 | 50.11 | 72.36 | 59.35 | 56.17 |
| | | | GPTAQ | 11.61 | 11.98 | 37.8 | 67.47 | 53.33 | 74.76 | 65.04 | 59.68 |
| | | | MagR | 11.00 | 11.22 | 44.37 | 75.88 | 56.58 | 77.31 | 71.59 | 65.15 |
| | | | NeUQI | **6.63** | **9.97** | **51.88** | **81.61** | **59.23** | **79.87** | **75.69** | **69.66** |

*Table 13.* Results are reported for the LLaMA 2 family models under bfloat16, as well as for models quantized to 3-bit, 3-bit with group size 128, 4-bit, and 4-bit with group size 128. † denotes that we use bfloat16 for easier fine-tuning, whereas previous work uses float16. We omit the MagR result for LLaMA 2 13B under 3-bit quantization with group size 128 due to an anomalous result that deviates from the expected trend.

| Size | Bits | Group | Method | Wiki2↓ | C4↓ | ArcC↑ | ArcE↑ | HellaS↑ | PiQA↑ | WinoG↑ | Acc↑ |
|---|---|---|---|---|---|---|---|---|---|---|---|
| | BF16† | - | - | 5.12 | 6.63 | 43.34 | 76.26 | 57.17 | 77.97 | 68.98 | 64.74 |
| | | | GPTQ | 5.62 | 7.12 | 43.52 | 75.55 | 56.27 | 77.48 | **69.85** | **64.53** |
| | | 128 | GPTAQ | 5.61 | 7.10 | **43.60** | **75.63** | **56.71** | 77.48 | 68.75 | 64.43 |
| | | | MagR | 5.69 | 7.15 | 41.64 | 75.42 | 55.57 | 77.15 | 68.98 | 63.75 |
| | 4 | | NeUQI | **5.60** | **7.09** | 42.92 | 75.17 | 56.19 | **77.64** | 69.46 | 64.27 |
| | | | GPTQ | 5.84 | 7.36 | 41.64 | 74.16 | 55.84 | **77.69** | **69.85** | **63.84** |
| | | - | GPTAQ | 5.79 | 7.31 | 41.30 | **75.38** | **55.91** | 77.20 | 68.82 | 63.72 |
| | | | MagR | **5.75** | **7.25** | **42.32** | 74.62 | 55.31 | 77.53 | 67.88 | 63.53 |
| 7B | | | NeUQI | 5.80 | 7.26 | 41.89 | 74.87 | 54.78 | 76.77 | 68.51 | 63.36 |
| | | | GPTQ | 6.34 | 7.86 | 39.76 | **73.74** | 53.91 | **77.31** | 67.48 | 62.44 |
| | | 128 | GPTAQ | 6.18 | 7.73 | **40.53** | 73.65 | **54.71** | 76.88 | **68.43** | **62.84** |
| | | | MagR | 6.27 | 7.76 | 40.10 | 72.01 | 53.04 | **77.31** | 67.09 | 61.91 |
| | 3 | | NeUQI | **6.07** | **7.58** | 38.57 | 72.35 | 54.19 | 76.44 | 67.48 | 61.81 |
| | | | GPTQ | 8.45 | 9.87 | 34.73 | 66.46 | 49.08 | 73.34 | 64.40 | 57.60 |
| | | - | GPTAQ | 7.81 | 9.33 | 35.92 | 68.86 | 50.85 | 74.27 | 65.98 | 59.17 |
| | | | MagR | 6.65 | 8.22 | **38.91** | **72.01** | **52.32** | 75.03 | 66.93 | **61.04** |
| | | | NeUQI | **6.56** | **8.10** | 37.97 | 71.42 | 50.75 | **75.41** | **68.19** | 60.75 |
| | BF16† | - | - | 4.57 | 6.05 | 48.21 | 79.46 | 60.09 | 79.11 | 72.30 | 67.83 |
| | | | GPTQ | 5.00 | 6.56 | **47.70** | 78.28 | 59.61 | 78.62 | 72.53 | 67.35 |
| | | 128 | GPTAQ | **4.98** | **6.55** | 47.44 | 78.45 | **59.82** | 78.89 | 71.82 | 67.28 |
| | | | MagR | 5.03 | 6.59 | 46.59 | 78.66 | 59.21 | 78.73 | **73.16** | 67.27 |
| | 4 | | NeUQI | **4.98** | 6.56 | 47.10 | **79.21** | 59.55 | **79.11** | 72.30 | **67.45** |
| | | | GPTQ | 5.15 | 6.71 | 44.97 | 77.44 | 58.88 | 77.91 | 71.27 | 66.09 |
| | | - | GPTAQ | 5.13 | 6.69 | **45.48** | 77.53 | **59.25** | **78.24** | 71.03 | 66.31 |
| | | | MagR | **5.09** | **6.65** | 45.31 | **78.54** | 59.02 | **78.24** | **71.90** | **66.60** |
| 13B | | | NeUQI | **5.09** | 6.67 | 45.31 | 77.86 | 58.64 | 78.13 | 71.19 | 66.23 |
| | | | GPTQ | 5.43 | 7.05 | 45.22 | 77.23 | 58.14 | 77.69 | 70.72 | 65.80 |
| | | 128 | GPTAQ | 5.38 | 6.96 | **46.50** | **78.37** | **58.54** | 78.35 | 70.64 | 66.48 |
| | | | MagR | - | - | - | - | - | - | - | - |
| | 3 | | NeUQI | **5.32** | **6.91** | 45.82 | 78.32 | 57.79 | **78.40** | **72.77** | **66.62** |
| | | | GPTQ | 6.46 | 8.03 | 38.91 | 73.48 | 55.18 | 76.39 | 68.35 | 62.46 |
| | | - | GPTAQ | 6.36 | 7.89 | 40.87 | 72.85 | 55.35 | 77.26 | 68.27 | 62.92 |
| | | | MagR | 5.74 | 7.32 | 42.32 | **76.56** | **56.56** | **77.80** | 69.38 | 64.52 |
| | | | NeUQI | **5.70** | **7.25** | **42.75** | 75.84 | 56.06 | 77.64 | **70.72** | **64.60** |
| | BF16† | - | - | 3.12 | 4.97 | 54.52 | 82.66 | 64.76 | 82.15 | 77.43 | 72.30 |
| | | | GPTQ | 3.42 | **5.58** | 54.69 | 82.74 | **64.45** | 81.83 | 77.19 | 72.18 |
| | | 128 | GPTAQ | 3.42 | **5.58** | 54.18 | 82.53 | 64.38 | 81.66 | 76.56 | 71.86 |
| | | | MagR | 3.46 | 5.61 | 54.44 | 82.53 | 64.03 | 81.66 | 77.35 | 72.00 |
| | 4 | | NeUQI | **3.41** | **5.58** | **54.95** | **82.83** | 64.29 | **82.05** | **77.98** | **72.42** |
| | | | GPTQ | 3.59 | 5.68 | 54.27 | 81.99 | **64.16** | 82.15 | 77.19 | 71.95 |
| | | - | GPTAQ | 3.58 | 5.68 | 53.84 | 82.37 | **64.16** | 81.50 | 76.56 | 71.69 |
| | | | MagR | 3.50 | 5.63 | 54.18 | 82.32 | 64.15 | **82.21** | 76.56 | 71.88 |
| 70B | | | NeUQI | **3.47** | **5.62** | **54.35** | **83.08** | 64.04 | 81.66 | **78.53** | **72.33** |
| | | | GPTQ | 3.87 | 5.86 | 53.24 | 81.57 | 63.16 | 81.61 | 77.43 | 71.40 |
| | | 128 | GPTAQ | 3.85 | 5.84 | 52.05 | 81.27 | **63.34** | 81.66 | **78.30** | 71.32 |
| | | | MagR | 3.81 | 5.82 | 53.75 | 82.32 | 62.95 | 81.50 | 77.19 | 71.54 |
| | 3 | | NeUQI | **3.71** | **5.77** | **54.95** | **82.45** | 62.99 | **81.99** | 77.03 | **71.88** |
| | | | GPTQ | 4.83 | 6.57 | 49.15 | 79.38 | 60.60 | 80.36 | 74.35 | 68.77 |
| | | - | GPTAQ | 4.78 | 6.51 | 49.83 | 80.77 | 61.24 | 80.96 | 74.98 | 69.56 |
| | | | MagR | 4.03 | 5.98 | 52.39 | 81.44 | 62.50 | 80.85 | **77.19** | 70.87 |
| | | | NeUQI | **3.90** | **5.90** | **53.50** | **83.00** | 62.58 | **81.61** | 76.72 | **71.48** |

*Table 14.* Results are reported for the LLaMA 3 family models under bfloat16, as well as for models quantized to 3-bit, 3-bit with group size 128, 4-bit, and 4-bit with group size 128.

| Size | Bits | Group | Method | Wiki2↓ | C4↓ | ArcC↑ | ArcE↑ | HellaS↑ | PiQA↑ | WinoG↑ | Acc↑ |
|------|------|-------|--------|--------|-----|-------|-------|---------|-------|--------|------|
| 8B | BF16 | - | - | 5.76 | 8.32 | 50.34 | 80.22 | 60.19 | 79.60 | 73.64 | 68.80 |
| | 4 | 128 | GPTQ | 6.19 | 8.99 | 47.61 | 77.86 | 59.06 | 77.75 | 73.88 | 67.23 |
| | | | GPTAQ | 6.53 | 9.36 | 48.98 | 79.25 | 59.41 | 78.67 | 73.09 | 67.88 |
| | | | MagR | 6.88 | 9.74 | 48.46 | 79.46 | 58.62 | **79.16** | 74.19 | 67.98 |
| | | | NeUQI | **6.12** | **8.87** | **50.00** | **80.18** | **59.53** | 78.67 | **74.90** | **68.66** |
| | | - | GPTQ | 6.97 | 9.95 | 44.54 | 77.27 | 57.87 | 77.15 | 73.32 | 66.03 |
| | | | GPTAQ | 7.16 | 10.16 | 43.17 | 75.72 | **58.23** | 77.04 | 72.38 | 65.31 |
| | | | MagR | 7.65 | 10.04 | **47.10** | 76.89 | 57.73 | 77.64 | 72.77 | 66.43 |
| | | | NeUQI | **6.67** | **9.41** | **47.10** | **77.69** | 58.13 | **79.54** | **74.59** | **67.41** |
| | 3 | 128 | GPTQ | 8.30 | 11.50 | 40.19 | 71.84 | 54.43 | 76.22 | 70.72 | 62.68 |
| | | | GPTAQ | 8.34 | 11.70 | 42.32 | 73.65 | **56.05** | 77.26 | 72.06 | 64.27 |
| | | | MagR | 8.63 | 11.42 | 39.08 | 73.61 | 52.66 | 75.73 | 72.77 | 62.77 |
| | | | NeUQI | **7.45** | **10.49** | **46.50** | **79.34** | 55.28 | **77.69** | **73.48** | **66.46** |
| | | - | GPTQ | 19.03 | 29.26 | 25.34 | 46.25 | 43.02 | 62.84 | 59.91 | 47.47 |
| | | | GPTAQ | 14.39 | 18.40 | 28.58 | 53.32 | 45.82 | 68.28 | 64.72 | 52.15 |
| | | | MagR | 9.83 | 12.70 | 40.53 | 71.76 | 53.61 | 76.39 | 70.09 | 62.48 |
| | | | NeUQI | **9.70** | **11.61** | **43.77** | **76.30** | **53.87** | **77.20** | **71.35** | **64.50** |
| 70B | BF16 | - | - | 2.68 | 5.88 | 60.49 | 86.95 | 66.37 | 82.48 | 80.90 | 75.44 |
| | 4 | 128 | GPTQ | 3.40 | 6.41 | 57.76 | **85.14** | **66.00** | 82.05 | 80.03 | 74.20 |
| | | | GPTAQ | 3.39 | 7.04 | 57.85 | **85.14** | 65.94 | 81.88 | 79.01 | 73.96 |
| | | | MagR | 3.75 | 7.33 | 56.83 | 84.72 | 65.31 | 81.72 | 80.43 | 73.80 |
| | | | NeUQI | **3.17** | **6.25** | **58.02** | 84.76 | 65.82 | **82.43** | **80.90** | **74.39** |
| | | - | GPTQ | 1486 | 1404 | 19.97 | 25.21 | 30.47 | 52.94 | 50.83 | 35.88 |
| | | | GPTAQ | >1e4 | 5169 | 20.65 | 25.97 | 25.92 | 53.26 | 48.62 | 34.88 |
| | | | MagR | 1856 | 1894 | 19.20 | 28.16 | 28.77 | 54.35 | 50.67 | 36.23 |
| | | | NeUQI | **4.90** | **10.00** | **51.96** | **80.47** | **62.29** | **79.49** | **63.61** | **67.57** |
| | 3 | 128 | GPTQ | 5.30 | 8.33 | 51.37 | 80.93 | 62.73 | **80.90** | 75.53 | 70.29 |
| | | | GPTAQ | 5.77 | 8.42 | 54.35 | 82.83 | **63.80** | 80.69 | 73.95 | 71.12 |
| | | | MagR | 5.21 | 8.16 | **55.20** | **83.67** | 63.56 | 80.63 | 78.85 | **72.38** |
| | | | NeUQI | **4.63** | **7.58** | 54.52 | 83.42 | 62.50 | 80.47 | **80.03** | 72.19 |
| | | - | GPTQ | 2645 | 1111 | 20.82 | 25.04 | 26.16 | 52.18 | 51.22 | 35.08 |
| | | | GPTAQ | >1e4 | 7797 | 20.65 | 25.72 | 25.82 | 52.72 | 51.54 | 35.29 |
| | | | MagR | 1621 | 1087 | 20.22 | 25.17 | 26.17 | 52.50 | 50.36 | 34.88 |
| | | | NeUQI | **9.04** | **13.36** | **35.84** | **69.11** | **52.97** | **71.82** | **55.80** | **57.11** |

*Table 15.* Results are reported for the Qwen 2.5 7B, 14B, and 32B models under bfloat16, as well as for models quantized to 3-bit, 3-bit with group size 128, 4-bit, and 4-bit with group size 128.

| Size | Bits | Group | Method | Wiki2↓ | C4↓ | ArcC↑ | ArcE↑ | HellaS↑ | PiQA↑ | WinoG↑ | Acc↑ |
|------|------|-------|--------|--------|-----|-------|-------|---------|-------|--------|------|
| 7B | BF16 | - | - | 6.39 | 10.02 | 48.29 | 80.56 | 60.00 | 78.67 | 72.69 | 68.04 |
| | 4 | 128 | GPTQ | 6.61 | 10.19 | 48.21 | 80.35 | 59.24 | **79.05** | 71.35 | 67.64 |
| | | | GPTAQ | 7.10 | 10.62 | 48.63 | 79.88 | 59.32 | 78.73 | 71.67 | 67.65 |
| | | | MagR | 7.21 | 10.82 | **51.45** | 80.72 | 58.78 | 78.67 | 72.45 | **68.42** |
| | | | NeUQI | **6.55** | **10.17** | 48.63 | **80.98** | **59.36** | 78.67 | **73.95** | 68.32 |
| | | - | GPTQ | 7.06 | 10.61 | 46.59 | 79.25 | 58.08 | 78.56 | 68.59 | 66.21 |
| | | | GPTAQ | 7.58 | 11.05 | 46.84 | **80.39** | 58.47 | **78.89** | 69.53 | 66.83 |
| | | | MagR | 7.39 | 10.96 | 45.39 | 76.05 | 58.49 | 78.51 | **72.61** | 66.21 |
| | | | NeUQI | **6.82** | **10.42** | **48.55** | 79.38 | **58.63** | 78.78 | 71.90 | **67.45** |
| | 3 | 128 | GPTQ | 7.42 | 10.94 | 43.77 | 75.63 | 56.71 | 77.37 | 65.75 | 63.85 |
| | | | GPTAQ | 7.97 | 11.38 | 44.71 | 77.27 | **57.17** | **78.51** | 69.22 | 65.38 |
| | | | MagR | 7.92 | 11.43 | 45.73 | 77.69 | 56.81 | 78.24 | **71.11** | 65.92 |
| | | | NeUQI | **7.14** | **10.80** | **46.42** | **80.05** | 56.46 | 78.35 | 70.40 | **66.34** |
| | | - | GPTQ | 10.73 | 13.90 | 39.33 | 71.21 | 50.16 | 73.29 | 60.77 | 58.95 |
| | | | GPTAQ | 11.53 | 14.22 | 35.92 | 64.94 | 51.42 | 75.03 | 61.72 | 57.81 |
| | | | MagR | 8.66 | 12.12 | **45.14** | 74.41 | **55.60** | 77.48 | 66.22 | 63.77 |
| | | | NeUQI | **8.64** | **11.72** | 44.80 | **76.77** | 55.15 | **77.97** | **70.96** | **65.13** |
| 14B | BF16 | - | - | 4.93 | 8.75 | 55.80 | 82.37 | 63.38 | 81.07 | 74.66 | 71.46 |
| | 4 | 128 | GPTQ | 5.29 | 8.93 | 54.35 | 81.94 | 62.94 | 80.96 | 74.98 | 71.03 |
| | | | GPTAQ | 5.70 | 9.32 | 53.33 | 82.03 | **62.95** | 81.01 | 74.74 | 70.81 |
| | | | MagR | 5.71 | 9.38 | 55.12 | 81.44 | 62.61 | **81.12** | **77.11** | 71.48 |
| | | | NeUQI | **5.20** | **8.89** | **56.40** | **83.25** | 62.78 | 81.01 | 75.61 | **71.81** |
| | | - | GPTQ | 5.80 | 9.24 | 52.73 | 81.44 | 62.27 | 80.20 | 74.98 | 70.32 |
| | | | GPTAQ | 6.23 | 9.62 | **54.61** | **82.62** | 61.90 | 80.25 | 73.40 | 70.56 |
| | | | MagR | 5.96 | 9.49 | **54.61** | 81.82 | 62.52 | **80.36** | 75.53 | **70.97** |
| | | | NeUQI | **5.46** | **9.05** | 52.56 | 81.10 | **62.62** | **80.36** | **77.19** | 70.77 |
| | 3 | 128 | GPTQ | 6.29 | 9.63 | 48.55 | 79.38 | 60.06 | 79.16 | 72.14 | 67.86 |
| | | | GPTAQ | 6.82 | 10.00 | 48.38 | 79.21 | 60.68 | 79.33 | 72.45 | 68.01 |
| | | | MagR | 6.58 | 9.91 | 51.88 | 80.22 | **60.83** | 79.82 | 75.45 | 69.64 |
| | | | NeUQI | **5.91** | **9.42** | **52.90** | **81.02** | 60.71 | **80.90** | **76.72** | **70.45** |
| | | - | GPTQ | 8.48 | 11.61 | 38.82 | 67.21 | 54.22 | 75.52 | 65.51 | 60.26 |
| | | | GPTAQ | 9.36 | 11.97 | 42.24 | 71.21 | 55.45 | 77.20 | 64.09 | 62.04 |
| | | | MagR | 7.34 | 10.41 | 48.81 | 79.97 | 59.32 | **79.71** | 71.43 | 67.85 |
| | | | NeUQI | **6.82** | **9.89** | **50.17** | **80.43** | **59.39** | 79.54 | **76.09** | **69.12** |
| 32B | BF16 | - | - | 4.67 | 8.59 | 53.16 | 80.51 | 65.00 | 81.99 | 75.69 | 71.27 |
| | 4 | 128 | GPTQ | 4.87 | 8.69 | **54.10** | **81.44** | 64.59 | 80.69 | 75.93 | **71.35** |
| | | | GPTAQ | 5.25 | 9.06 | **54.10** | 81.40 | 64.41 | 80.74 | 75.61 | 71.25 |
| | | | MagR | 5.24 | 9.08 | 52.47 | 80.89 | 64.22 | **81.72** | **76.80** | 71.22 |
| | | | NeUQI | **4.82** | **8.68** | 53.24 | 80.26 | **64.71** | 81.18 | 76.64 | 71.20 |
| | | - | GPTQ | 5.23 | 8.89 | 51.19 | 79.29 | 63.96 | 80.41 | 75.45 | 70.06 |
| | | | GPTAQ | 5.65 | 9.26 | 51.19 | 79.29 | 63.81 | 80.69 | 74.19 | 69.84 |
| | | | MagR | 5.45 | 9.17 | **53.67** | 81.48 | 63.89 | 81.61 | 74.90 | 71.11 |
| | | | NeUQI | **4.98** | **8.77** | 53.07 | **81.48** | **64.46** | **81.77** | **77.35** | **71.63** |
| | 3 | 128 | GPTQ | 5.63 | 9.11 | 51.11 | 79.50 | 62.83 | 80.36 | 75.14 | 69.79 |
| | | | GPTAQ | 6.11 | 9.47 | 50.51 | 80.47 | 62.84 | 80.30 | 72.45 | 69.32 |
| | | | MagR | 5.87 | 9.41 | 51.28 | 79.84 | **63.32** | **81.07** | 75.14 | 70.13 |
| | | | NeUQI | **5.30** | **8.99** | **51.71** | **82.07** | 62.92 | 80.90 | **76.40** | **70.80** |
| | | - | GPTQ | 7.21 | 10.36 | 44.03 | 74.96 | 58.75 | 77.97 | 66.22 | 64.38 |
| | | | GPTAQ | 7.92 | 10.78 | 43.52 | 73.74 | 59.31 | 78.13 | 67.40 | 64.42 |
| | | | MagR | 6.52 | 9.82 | **51.19** | 80.47 | 61.71 | **80.85** | 73.09 | 69.46 |
| | | | NeUQI | **5.85** | **9.31** | **51.19** | **80.98** | **62.30** | 80.14 | **75.14** | **69.95** |

*Table 16.* Results are reported for the Qwen 2.5 72B models under bfloat16, as well as for models quantized to 3-bit, 3-bit with group size 128, 4-bit, and 4-bit with group size 128.

| Size | Bits | Group | Method | Wiki2↓ | C4↓ | ArcC↑ | ArcE↑ | HellaS↑ | PiQA↑ | WinoG↑ | Acc↑ |
|------|------|-------|--------|--------|-----|-------|-------|---------|-------|--------|------|
| 72B | BF16 | - | - | 3.64 | 7.75 | 58.11 | 84.76 | 67.59 | 82.10 | 77.35 | 73.98 |
| | 4 | 128 | GPTQ | 3.82 | 7.85 | **57.51** | 84.60 | 67.14 | 82.26 | 78.06 | 73.91 |
| | | | GPTAQ | 4.08 | 8.26 | **57.51** | **84.93** | **67.38** | **82.43** | 78.53 | **74.16** |
| | | | MagR | 4.15 | 8.34 | 56.40 | 84.89 | 67.04 | 82.37 | 78.30 | 73.80 |
| | | | NeUQI | **3.78** | **7.83** | **57.51** | 84.47 | 67.16 | 81.99 | **78.93** | 74.01 |
| | | - | GPTQ | 4.17 | 8.35 | 55.38 | 84.68 | 66.68 | 81.66 | **77.90** | 73.26 |
| | | | GPTAQ | 4.60 | 8.45 | 57.42 | 84.64 | 66.75 | 81.88 | 76.16 | 73.37 |
| | | | MagR | 4.59 | 8.37 | 56.83 | **84.85** | 66.76 | **82.15** | 77.66 | 73.65 |
| | | | NeUQI | **3.95** | **7.91** | **58.02** | 84.34 | **66.81** | **82.15** | **77.90** | **73.85** |
| | 3 | 128 | GPTQ | 4.55 | 8.53 | **55.97** | **83.92** | 65.66 | 80.79 | 77.74 | 72.82 |
| | | | GPTAQ | 5.01 | 8.63 | 55.72 | 83.38 | 65.75 | 81.50 | 77.11 | 72.69 |
| | | | MagR | 4.93 | 8.62 | 53.84 | 83.29 | **66.12** | 82.10 | **79.32** | **72.93** |
| | | | NeUQI | **4.26** | **8.11** | 54.52 | 83.75 | 65.85 | **82.21** | 78.06 | 72.88 |
| | | - | GPTQ | 6.89 | 9.96 | 46.93 | 78.03 | 61.51 | 78.62 | 71.43 | 67.30 |
| | | | GPTAQ | 8.15 | 9.81 | 44.03 | 76.39 | 61.71 | 79.16 | 70.01 | 66.26 |
| | | | MagR | 5.34 | 8.90 | 55.72 | 83.00 | **64.97** | 81.39 | 75.06 | 72.03 |
| | | | NeUQI | **4.99** | **8.37** | **56.40** | **83.75** | 64.92 | **81.83** | **78.45** | **73.07** |

