# OpenReview forum: "NeUQI: Near-Optimal Uniform Quantization Parameter Initialization for Low-Bit LLMs"
_ICML.cc/2026/Conference — ICML 2026 regular_

### Official Review · Reviewer_Wbub · 2026-02-21

**Soundness:** 2
**Presentation:** 2
**Significance:** 2
**Originality:** 2
**Overall Recommendation:** 4
**Confidence:** 4

**Summary:**

This work proposes a method for LLM quantization via joint scale / zero point optimization leveraging diagonal Hessian approximation. The introduced approach is validated on a compression of several language model families (Llama-⅔, Qwen-2.5). The NeUQI method is shown to be coupled with transforms and post-hoc distillation.

**Compliance With Llm Reviewing Policy:**

Affirmed.

**Final Justification:**

After reading the responses addressed to me and other reviewers I am leaning towards acceptance.
The empirical evidence justifying the efficacy of the method is sufficient.

**Key Questions For Authors:**

- How does the method perform compared with GPTQ using optimal MSE scales [1]?
- What is the total runtime cost of NeUQI quantization for a model like Llama-2-7B or Llama-3-8B?
- Can the proposed scale/zero-point strategy be integrated into GPTQ?
---
References

[1] https://github.com/IST-DASLab/marlin

**Limitations:**

-

**Strengths And Weaknesses:**

Strengths

- The proposed method despite simple shows decent performance outperforming baselines without scale/zero-point tuning.

Weaknesses

- The idea of scale/zero-point optimization does not appear to be novel. HQQ [1] performs scale/zero-point optimization in the data-free setting; however, it is neither discussed nor compared against, despite being a natural baseline.
- Baselines such as OmniQuant and HQQ are not included in Table 2 (the main comparison table).
- The results reported for the PV-Tuning baseline look suspicious. Specifically, the original paper reports a perplexity (ppl) of 5.84 and an average accuracy of 61.35, whereas the paper reports 8.49 and 52.17. Given the high tuning cost, it is reasonable not to compare against PV-Tuning, but the discrepancy should be explained.

---
References

[1] https://dropbox.github.io/hqq_blog/

[2] Malinovskii, Vladimir, et al. "Pv-tuning: Beyond straight-through estimation for extreme llm compression." Advances in Neural Information Processing Systems 37 (2024): 5074-5121.

---

> ### Author Rebuttal · Authors · 2026-03-28
>
> Thank you for your valuable comments and suggestions. Our detailed responses to each concern are given below.
>
> > W1: The idea of scale/zero-point optimization does not appear to be novel. HQQ [1] performs scale/zero-point optimization in the data-free setting; however, it is neither discussed nor compared against, despite being a natural baseline.
>
> HQQ yields perplexity values above 100 for the LLaMA 2 family under 2-bit, 2-bit with group size 128, and 3-bit quantization settings. These results suggest that optimizing only the zero-point, as in HQQ, is insufficient to preserve model quality.
>
> Furthermore, the HQQ blog reports 2-bit results for the LLaMA 2 family using group sizes of 32 and 16. In practice, the effective bit-width is 3 and 4 bits per weight, respectively, which is not a fair setting for evaluating true low-bit quantization.
>
>
> > W2: Baselines such as OmniQuant and HQQ are not included in Table 2 (the main comparison table).
>
> For OmniQuant, it relies on fine-tuning and is therefore not directly comparable to methods without fine-tuning. Accordingly, it is compared only with other fine-tuning-based approaches, with the results reported in Table 6.
>
> For HQQ, as discussed in our response to W1, its performance is relatively weak.
>
> > W3: The results reported for the PV-Tuning baseline look suspicious. Specifically, the original paper reports a perplexity (ppl) of 5.84 and an average accuracy of 61.35, whereas the paper reports 8.49 and 52.17. Given the high tuning cost, it is reasonable not to compare against PV-Tuning, but the discrepancy should be explained.
>
> We focus on uniform quantization, whereas the result with perplexity 5.84 in Table 2 of the PV-Tuning paper is based on AQLM, which employs vector quantization and thus lies outside the scope of our study. For a fair comparison under the same quantization scheme, we instead refer to the GPTQ results with uniform quantization reported in Table 1 of the same paper.
>
> > Q1: How does the method perform compared with GPTQ using optimal MSE scales [1]?
>
> We provide the comparison results for 2-bit quantization as follows. Compared to activation-aware methods, MSE exhibits a significant performance gap. These results will be incorporated into Table 4 in the final version.
>
> | Model            | Method     | Wiki2 | C4    | Acc   |
> |------------------|------------|-------|-------|-------|
> | LLaMA 2 7B       | MSE        | 65.42 | 93.58 | 44.19 |
> |                  | Int-Search | 26.26 | 24.15 | 43.85 |
> |                  | NeUQI      | 17.14 | 17.50 | 47.24 |
> | LLaMA 2 13B      | MSE        | 155   | 223   | 36.42 |
> |                  | Int-Search | 15.67 | 16.69 | 47.60 |
> |                  | NeUQI      | 13.72 | 14.39 | 48.76 |
> | LLaMA 2 70B      | MSE        | 11.98 | 13.16 | 58.10 |
> |                  | Int-Search | 8.71  | 10.49 | 61.61 |
> |                  | NeUQI      | 7.03  | 8.88  | 65.13 |
> | LLaMA 3 8B       | MSE        | 147   | 57.94 | 37.82 |
> |                  | Int-Search | 93.64 | 47.76 | 40.78 |
> |                  | NeUQI      | 64.47 | 39.41 | 47.30 |
>
>
> > Q2: What is the total runtime cost of NeUQI quantization for a model like Llama-2-7B or Llama-3-8B?
>
> The offline quantization runtime for the LLaMA 2 family is reported in Appendix F.
>
> > Q3: Can the proposed scale/zero-point strategy be integrated into GPTQ?
>
> Yes. Our method handles quantization parameter initialization, while GPTQ performs the subsequent weight matrix adjustment. Our method is orthogonal to GPTQ and can be seamlessly integrated into it.

---

> > ### Author Rebuttal · Reviewer_Wbub · 2026-04-01
> >
> > Most of my concerns were resolved, therefore I decided to raise my score.

---

> > > ### Author Response · Authors · 2026-04-02
> > >
> > > We sincerely appreciate your assessment and are grateful for the time and effort you devoted to evaluating our paper. We are encouraged that our clarifications helped address your concerns. If accepted, we will ensure that all additional experiments introduced during the rebuttal are fully incorporated into the final version.

---

### Official Review · Reviewer_K4qr · 2026-03-05

**Soundness:** 3
**Presentation:** 3
**Significance:** 3
**Originality:** 4
**Overall Recommendation:** 5
**Confidence:** 4

**Summary:**

This paper introduces NeUQI, a novel approach for quickly finding near-optimal scale and zero-point parameters in post-training quantization (PTQ). By temporarily fixing the scale, the method efficiently computes a near-optimal zero-point, reducing the complex joint search to a much simpler single-variable optimization. The two core contributions of this work are: 1) Identifying Limitations: Revealing the two previously overlooked constraints of the traditional Min-Max formula, which restrict optimization. 2) Algorithmic Efficiency: Developing a fast zero-point search algorithm that operates in $O(n \log n)$ time by strategically reducing transition points. Ultimately, NeUQI delivers a significant performance advantage over recent baselines, particularly in the highly challenging 2-bit quantization setting.

**Compliance With Llm Reviewing Policy:**

Affirmed.

**Final Justification:**

The response has resolved my questions. I keep the current score.

**Key Questions For Authors:**

1. Can you explain more about the lightweight distillation in your paper?
2. Can you share the results of GPTAQ and MagR for 3 bit and 4 bit in table 12, table 13 and table 14?

**Limitations:**

yes

**Strengths And Weaknesses:**

Strengths:
1. Overcoming Traditional Limits: Reveals two previously overlooked constraints of the traditional Min-Max formula and successfully applies these insights to advance non-fine-tuning quantization methods.
2. Algorithmic Efficiency: Introduces a novel and highly efficient zero-point search algorithm that significantly reduces computational complexity.
3. Superior Performance: Demonstrates that the proposed approach significantly outperforms existing baselines, particularly in the highly challenging 2-bit quantization setting.
Weaknesses:
1. Table 12, table 13 and table 14 do not contain GPTAQ and MagR for 3 bit and 4 bit.
2. Although the proposed approach significantly outperforms existing baselines in the 2-bit setting, a noticeable performance gap remains when compared to full-precision bfloat16 models. Consequently, the practical application value of 2-bit quantization is still relatively limited compared to more stable 3-bit or 4-bit configurations.

---

> ### Author Rebuttal · Authors · 2026-03-30
>
> Thank you for your valuable comments and suggestions. Our detailed responses to each concern are given below.
>
> > W1: Table 12, table 13 and table 14 do not contain GPTAQ and MagR for 3 bit and 4 bit.
>
> We apologize for the omission. The results are presented below, and NeUQI still demonstrates clear advantages. Due to response length constraints, the results for the Qwen 2.5 family are omitted here and will be included in the final version. (There is an issue in the MagR experiments on LLaMA 2 13B, W3G128.)
>
> | Setting | Method | LLaMA 2 7B |  |  | LLaMA 2 13B |  |  | LLaMA 2 70B |  |  | LLaMA 3 8B |  |  | LLaMA 3 70B |  |  |
> |--------|--------|-----------|------|------|------------|------|------|------------|------|------|------------|------|------|-------------|------|------|
> |        |        | Wiki2 | C4 | Acc | Wiki2 | C4 | Acc | Wiki2 | C4 | Acc | Wiki2 | C4 | Acc | Wiki2 | C4 | Acc |
> | BF16 | - | 5.12 | 6.63 | 64.74 | 4.57 | 6.05 | 67.83 | 3.12 | 4.97 | 72.30 | 5.76 | 8.32 | 68.80 | 2.68 | 5.88 | 75.44 |
> | W3 | GPTQ | 8.45 | 9.87 | 57.60 | 6.46 | 8.03 | 62.46 | 4.83 | 6.57 | 68.77 | 19.03 | 29.26 | 47.47 | 2645.00 | 1111.00 | 35.08 |
> |  | GPTAQ | 7.81 | 9.33 | 59.17 | 6.36 | 7.89 | 62.92 | 4.78 | 6.51 | 69.56 | 14.39 | 18.40 | 52.15 | >1e4 | 7797.54 | 35.29 |
> |  | MagR | 6.65 | 8.22 | **61.04** | 5.74 | 7.28 | 64.57 | 4.03 | 5.98 | 70.87 | 9.83 | 12.70 | 62.48 | 1621.46 | 1087.82 | 34.88 |
> |  | NeUQI | **6.56** | **8.10** | 60.75 | **5.70** | **7.25** | **64.60** | **3.90** | **5.90** | **71.48** | **9.70** | **11.61** | **64.50** | **9.04** | **13.36** | **57.11** |
> | W3G128 | GPTQ | 6.34 | 7.86 | 62.44 | 5.43 | 7.05 | 65.80 | 3.87 | 5.86 | 71.40 | 8.30 | 11.50 | 62.68 | 5.30 | 8.33 | 70.29 |
> |  | GPTAQ | 6.18 | 7.73 | **62.84** | 5.38 | 6.96 | 66.48 | 3.85 | 5.84 | 71.32 | 8.34 | 11.70 | 64.27 | 5.77 | 8.42 | 71.12 |
> |  | MagR | 6.25 | 7.76 | 62.72 | - | - | - | 3.81 | 5.82 | 71.54 | 8.63 | 11.42 | 62.77 | 5.21 | 8.16 | **72.38** |
> |  | NeUQI | **6.07** | **7.58** | 61.81 | **5.32** | **6.91** | **66.62** | **3.71** | **5.77** | **71.88** | **7.45** | **10.49** | **66.46** | **4.63** | **7.58** | 72.19 |
> | W4 | GPTQ | 5.84 | 7.36 | **63.84** | 5.15 | 6.71 | 66.09 | 3.59 | 5.68 | 71.95 | 6.97 | 9.95 | 66.03 | 1486.00 | 1404.00 | 35.88 |
> |  | GPTAQ | 5.79 | 7.31 | 63.72 | 5.13 | 6.69 | 66.31 | 3.58 | 5.68 | 71.69 | 7.16 | 10.16 | 65.31 | >1e4 | 5169.32 | 34.88 |
> |  | MagR | **5.75** | **7.25** | 63.53 | **5.09** | **6.65** | **66.66** | 3.50 | 5.63 | 71.88 | 7.65 | 10.04 | 66.43 | 1856.19 | 1894.26 | 36.23 |
> |  | NeUQI | 5.80 | 7.26 | 63.36 | **5.09** | 6.67 | 66.23 | **3.47** | **5.62** | **72.33** | **6.67** | **9.41** | **67.41** | **4.90** | **10.00** | **67.57** |
> | W4G128 | GPTQ | 5.62 | 7.12 | **64.53** | 5.00 | 6.56 | 67.35 | 3.42 | **5.58** | 72.18 | 6.19 | 8.99 | 67.23 | 3.40 | 6.41 | 74.20 |
> |  | GPTAQ | 5.61 | 7.10 | 64.43 | **4.98** | **6.55** | 67.28 | 3.42 | **5.58** | 71.86 | 6.53 | 9.36 | 67.88 | 3.39 | 7.04 | 73.96 |
> |  | MagR | 5.69 | 7.15 | 63.75 | 5.03 | 6.59 | 67.36 | 3.46 | 5.61 | 72.00 | 6.88 | 9.74 | 67.98 | 3.75 | 7.33 | 73.80 |
> |  | NeUQI | **5.60** | **7.09** | 64.27 | **4.98** | 6.56 | **67.45** | **3.41** | **5.58** | **72.42** | **6.12** | **8.87** | **68.66** | **3.17** | **6.25** | **74.39** |
>
> > W2: Although the proposed approach significantly outperforms existing baselines in the 2-bit setting, a noticeable performance gap remains when compared to full-precision bfloat16 models. Consequently, the practical application value of 2-bit quantization is still relatively limited compared to more stable 3-bit or 4-bit configurations.
>
> We agree that the 2-bit setting exhibits a larger performance gap compared to 3-bit and 4-bit configurations, due to its inherently limited representational capacity. The goal of our work is to advance the practicality of low-bit (e.g., 2-bit) uniform quantization by addressing the specific issue of quantization parameter initialization. Achieving optimal performance for 2-bit uniform quantization remains an open problem and is beyond the scope of our work. We believe this direction merits further investigation in future research.
>
>
> > Q1: Can you explain more about the lightweight distillation in your paper?
>
> In the lightweight distillation setting, all continuous parameters in the quantized model, including RMSNorm weights and quantization parameters such as scale and zero-point, are updated, while those in the embedding layer (embed_tokens) and the language model head (lm_head) are kept fixed. The loss function is defined as the mean squared error (MSE) between the output hidden states of the final transformer block of the full-precision and quantized models. Additional implementation details are provided in Appendix D.1.2.
>
> > Q2: Can you share the results of GPTAQ and MagR for 3 bit and 4 bit in table 12, table 13 and table 14?
>
> See W1

---

> > ### Author Rebuttal · Reviewer_K4qr · 2026-04-01
> >
> > Thank you for your response. Your answers have addressed my concerns, and I will retain my current score.

---

> > > ### Author Response · Authors · 2026-04-02
> > >
> > > We sincerely appreciate your assessment and are grateful for the time and effort you devoted to evaluating our paper. We are encouraged that our clarifications helped address your concerns. If accepted, we will ensure that all additional experiments introduced during the rebuttal are fully incorporated into the final version.

---

### Official Review · Reviewer_nqwU · 2026-03-09

**Soundness:** 3
**Presentation:** 3
**Significance:** 3
**Originality:** 2
**Overall Recommendation:** 4
**Confidence:** 5

**Summary:**

This paper investigates the impact of using floating-point zero-points (as opposed to integer zero-points) during uniform model quantization. To achieve better quantization performance, the authors propose a novel zero-point initialization method that combines analytical solving with a search mechanism. A series of experiments across various quantization configurations are conducted to demonstrate the effectiveness of the proposed approach.

**Compliance With Llm Reviewing Policy:**

Affirmed.

**Final Justification:**

The authors have provided beneficial evidence showing that proper zero-point initialization remains effective under multiple quantization frameworks. Consequently, I am raising my score.

**Key Questions For Authors:**

1. Ablation on Efficient-QAT Initialization: I request an ablation study comparing the proposed pipeline against a baseline that simply introduces a floating-point zero-point using Efficient-QAT's native initialization and fine-tuning mechanisms.

2. If one simply sets the zero-point as a trainable parameter within a standard layer-wise distillation framework (using a standard asymmetric quantization configuration), would the performance improve similarly? The authors need to clarify the advantage of their specific design over a naive trainable zero-point approach.

3. The performance of the GPTQ+Hadamard configuration in the 2-bit setting is abnormally low. Could the authors clarify the exact quantization grid used here? Specifically, I suspect that if the quantization points are set to {-2, -1, 0, 1} and the min/max values are clipped to 1, the actual number of usable quantization bins is reduced to 3. Please clarify this implementation detail and explain the anomalous drop in performance.

**Limitations:**

Yes

**Strengths And Weaknesses:**

Strengths:
1. Comprehensive Empirical Validation: The paper conducts extensive experiments across multiple baselines, clearly demonstrating that employing floating-point zero-points can yield noticeable improvements in quantization accuracy under various settings.

Weaknesses:
1.  The fundamental motivation of the paper is somewhat weak. It is a widely recognized and accepted fact in the quantization community (especially in practical deployment) that asymmetric quantization with floating-point zero-points improves model accuracy. The reason many prior academic works adopt symmetric quantization is not due to a lack of awareness of this fact, but rather to strip away complex quantization configurations. This allows them to isolate variables and highlight the specific algorithmic contributions of their proposed methods. Therefore, framing the use of floating-point zero-points as a major novel contribution significantly weakens the paper's standing.

2. The experimental results in Tables 2, 3, 4, and 5 appear to be built upon relatively weak baselines. While the proposed method shows incremental improvements over these baselines, the absolute performance—especially in the 2-bit setting—remains too poor to be practically usable. Demonstrating relative improvements on sub-optimal baselines does not strongly support the method's practical value.

3. Regarding the state-of-the-art results achieved by combining the proposed method with Efficient-QAT: While I understand that the authors apply their proposed asymmetric initialization prior to training, it remains unclear whether this specific initialization (analytical solving + search) is the true source of the performance gain. It is highly possible that simply applying Efficient-QAT's native initialization and fine-tuning strategies to a standard asymmetric floating-point zero-point scheme could achieve similar or even better results. Without isolating this, the necessity and contribution of the authors' specific initialization design remain questionable.

---

> ### Author Rebuttal · Authors · 2026-03-30
>
> Thank you for your valuable comments and suggestions. Our detailed responses to each concern are given below.
>
> > W1: The fundamental motivation of the paper is somewhat weak. It is a widely recognized and accepted fact in the quantization community (especially in practical deployment) that asymmetric quantization with floating-point zero-points improves model accuracy. The reason many prior academic works adopt symmetric quantization is not due to a lack of awareness of this fact, but rather to strip away complex quantization configurations. This allows them to isolate variables and highlight the specific algorithmic contributions of their proposed methods. Therefore, framing the use of floating-point zero-points as a major novel contribution significantly weakens the paper's standing.
>
> For 4-bit quantization, many representative works such as GPTQ adopt asymmetric quantization with integer zero-points rather than symmetric quantization. This can be observed from their use of Min-Max initialization instead of AbsMax initialization. Therefore, prior works are not limited to symmetric quantization as suggested.
>
> Moreover, the choice of symmetric quantization or asymmetric quantization with integer zero-points in many prior academic works is often inherited from earlier implementations or design conventions, rather than being the result of careful exploration of alternative configurations. In contrast, our work specifically focuses on this often overlooked aspect and investigates the optimization of quantization parameters, particularly scale and zero-point, for improved quantization performance.
>
> > W2: The experimental results in Tables 2, 3, 4, and 5 appear to be built upon relatively weak baselines. While the proposed method shows incremental improvements over these baselines, the absolute performance—especially in the 2-bit setting—remains too poor to be practically usable. Demonstrating relative improvements on sub-optimal baselines does not strongly support the method's practical value.
>
> We agree that methods without fine-tuning generally exhibit relatively weaker performance. However, fine-tuning-based approaches often introduce non-negligible computational overhead, which can be prohibitive in certain deployment scenarios. Therefore, evaluating non-fine-tuning methods remains practically meaningful.
> In this context, our results demonstrate that NeUQI consistently improves upon baseline methods, making it a more viable choice under such constraints. Furthermore, Tables 6 and 7 show that NeUQI can be effectively combined with fine-tuning and yields additional performance gains, indicating its applicability in both settings.
>
> > Q1: Ablation on Efficient-QAT Initialization: I request an ablation study comparing the proposed pipeline against a baseline that simply introduces a floating-point zero-point using Efficient-QAT's native initialization and fine-tuning mechanisms.
>
> For W3, Q1, and Q2, we apologize for the absence of an ablation study on EfficientQAT with a floating zero-point in our work. Table 6 in the EfficientQAT paper shows that simply introducing a floating-point zero-point leads to negligible performance changes. To directly address this concern, we re-evaluate LLaMA 2 7B under the 2-bit setting with a group size of 128. Our results confirm the same observation, as shown below.
>
> | Model | Method | Wiki2 ↓ | C4 ↓ | Acc ↑ |
> |-------|--------|---------|------|-------|
> | LLaMA 2 7B | EfficientQAT | 7.19 | 8.79 | 59.50 |
> | | + floating zero-point | 7.02 | 8.72 | 59.37 |
> | | + NeUQI | **6.81** | **8.41** | **60.01** |
> | LLaMA 3 8B | EfficientQAT | 9.80 | 13.22 | 59.37 |
> | | + floating zero-point | 9.72 | 13.22 | 59.22 |
> | | + NeUQI | **9.14** | **12.51** | **60.68** |
>
> > Q3: The performance of the GPTQ+Hadamard configuration in the 2-bit setting is abnormally low. Could the authors clarify the exact quantization grid used here? Specifically, I suspect that if the quantization points are set to {-2, -1, 0, 1} and the min/max values are clipped to 1, the actual number of usable quantization bins is reduced to 3. Please clarify this implementation detail and explain the anomalous drop in performance.
>
> We re-verified the results and confirm that the reported W2A16 GPTQ + Hadamard Transform corresponds to LLaMA 2 7B under 2-bit channel-wise quantization with Min–Max initialization + GPTQ + Hadamard Transform. The abnormally low performance, compared with results reported in other papers, is likely due to stronger or more advanced settings and algorithmic configurations than those used in our experiments.
>
> For the 2-bit setting, we adopt the quantization grid described in Eq. (1), namely {$s (i - z) | i \in {0,1,2,3} $}, where both $s\in \mathbb{R}$ and $z\in${0,1,2,3} represent the scale and zero-point, respectively.

---

> > ### Author Rebuttal · Reviewer_nqwU · 2026-04-01
> >
> > Thank you for your detailed rebuttal. While I still have some reservations regarding the limited novelty and simplicity of the method, your response has convinced me of the critical importance of a good zero-point initialization. I believe this is a beneficial finding that deserves to be shared with the community. Therefore, I have decided to raise my score.

---

> > > ### Author Response · Authors · 2026-04-02
> > >
> > > We sincerely appreciate your assessment and are grateful for the time and effort you devoted to evaluating our paper. We are encouraged that our response helped clarify the importance of a good zero-point initialization, and we appreciate your constructive comments regarding the method’s novelty and simplicity. If accepted, we will ensure that all additional experiments introduced during the rebuttal are fully incorporated into the final version.

---

### Official Review · Reviewer_WP2o · 2026-03-12

**Soundness:** 2
**Presentation:** 3
**Significance:** 3
**Originality:** 3
**Overall Recommendation:** 4
**Confidence:** 3

**Summary:**

Previous work on post-training quantizaiton mostly uses min-max initialization to determine the zero-point and the scale. However, min-max are easily affected by extreme values in the distribution. Therefore, this paper proposes NeUQI, which joinly optimizes the scale and zero-point. This not only provides a better post-training quantization quality, but also serves as a better initilization point if quantization aware training (QAT) is performed.

**Compliance With Llm Reviewing Policy:**

Affirmed.

**Final Justification:**

My concerns have been fully resolved. Thus I'd like to maintain my recommandation.

**Key Questions For Authors:**

The authors argue that NeUQI addresses the limitations of conventional Min-Max initialization, particularly its sensitivity to outliers. However, methods like SmoothQuant and QuIP already mitigate the impact of outliers by using channel-wise transformations or the Hadamard transform to 'smooth' the distribution and make it more amenable to quantization.
1. Since both approaches ultimately aim to reduce quantization error caused by extreme values, are NeUQI and these transformation-based methods essentially optimizing the same loss space?
2. If the weight distribution has already been smoothed—effectively 'fixing' the outliers—does the performance benefit of a near-optimal initialization like NeUQI diminish?
3. MagR explores a similar intersection in their work (e.g., Table 9), suggesting that distribution-level shifts can reduce the need for complex parameter search. How does NeUQI's contribution specifically remain distinct or additive in these cases?"

**Limitations:**

yes

**Strengths And Weaknesses:**

Strengths:

1. Identification of Min-Max Flaws: You are correct that the paper's core contribution is identifying two overlooked constraints in the standard Min-Max formula: the reliance on extreme values and the integer restriction on the zero-point.
2. Algorithmic and Theoretical 2.Depth: The paper provides a rigorous derivation for finding the global minimum of the piecewise quadratic loss function. It also includes Lemma B.1, which theoretically proves that their proposed "Min-Max+" adjustment is optimal for weights drawn from a uniform distribution.
3. Runtime Disclosure: The authors are transparent about the computational cost, providing a detailed breakdown in Table 10. They show that while NeUQI is slower than GPTQ, it is comparable to other search-based or magnitude-reduction methods.

Weaknesses:

1. The baselines are questionable. The main baselines that are used in this paper are GPTQ, GPTAQ, MagR. While GPTQ and GPTAQ are somewhat outdated, the comparision with the most relevant baseline, MagR is only included in 2-bit results, where the coordinate descent is removed for fair comparision. A more comprehensive experiments comparing NeUQI with MagR on 3- and 4-bit would greatly increases the strength of emprical results.
2. Missing State-of-the-Art (SOTA): The paper does not provide direct comparisons with QuIP or QuIP#, which are widely considered the benchmarks for ultra-low bit quantization. A direct comparision with these methods or an experiment suggested by question 3 would be helpful.

---

> ### Author Rebuttal · Authors · 2026-03-30
>
> Thank you for your valuable comments and suggestions. Our detailed responses to each concern are given below.
>
> > W1: The baselines are questionable. The main baselines that are used in this paper are GPTQ, GPTAQ, MagR. While GPTQ and GPTAQ are somewhat outdated, the comparision with the most relevant baseline, MagR is only included in 2-bit results, where the coordinate descent is removed for fair comparision. A more comprehensive experiments comparing NeUQI with MagR on 3- and 4-bit would greatly increases the strength of emprical results.
>
> GPTAQ and MagR serve as competitive baselines without fine-tuning or transformation-based methods. In addition, we include comparisons with stronger fine-tuning-based methods in Tables 6 and 7, and report results combined with the Hadamard Transformation in Table 3.
>
> We apologize for the omission. The results are presented below. Due to response length constraints, the results for the Qwen 2.5 family are omitted here and will be included in the final version. (There is an issue in the MagR experiments on LLaMA 2 13B, W3G128.)
>
> | Setting | Method | LLaMA 2 7B |  |  | LLaMA 2 13B |  |  | LLaMA 2 70B |  |  | LLaMA 3 8B |  |  | LLaMA 3 70B |  |  |
> |--------|--------|-----------|------|------|------------|------|------|------------|------|------|------------|------|------|-------------|------|------|
> |        |        | Wiki2 | C4 | Acc | Wiki2 | C4 | Acc | Wiki2 | C4 | Acc | Wiki2 | C4 | Acc | Wiki2 | C4 | Acc |
> | BF16 | - | 5.12 | 6.63 | 64.74 | 4.57 | 6.05 | 67.83 | 3.12 | 4.97 | 72.30 | 5.76 | 8.32 | 68.80 | 2.68 | 5.88 | 75.44 |
> | W3 | GPTQ | 8.45 | 9.87 | 57.60 | 6.46 | 8.03 | 62.46 | 4.83 | 6.57 | 68.77 | 19.03 | 29.26 | 47.47 | 2645.00 | 1111.00 | 35.08 |
> |  | MagR | 6.65 | 8.22 | **61.04** | 5.74 | 7.28 | 64.57 | 4.03 | 5.98 | 70.87 | 9.83 | 12.70 | 62.48 | 1621.46 | 1087.82 | 34.88 |
> |  | NeUQI | **6.56** | **8.10** | 60.75 | **5.70** | **7.25** | **64.60** | **3.90** | **5.90** | **71.48** | **9.70** | **11.61** | **64.50** | **9.04** | **13.36** | **57.11** |
> | W3G128 | GPTQ | 6.34 | 7.86 | 62.44 | 5.43 | 7.05 | 65.80 | 3.87 | 5.86 | 71.40 | 8.30 | 11.50 | 62.68 | 5.30 | 8.33 | 70.29 |
> |  | MagR | 6.25 | 7.76 | **62.72** | - | - | - | 3.81 | 5.82 | 71.54 | 8.63 | 11.42 | 62.77 | 5.21 | 8.16 | **72.38** |
> |  | NeUQI | **6.07** | **7.58** | 61.81 | **5.32** | **6.91** | **66.62** | **3.71** | **5.77** | **71.88** | **7.45** | **10.49** | **66.46** | **4.63** | **7.58** | 72.19 |
> | W4 | GPTQ | 5.84 | 7.36 | **63.84** | 5.15 | 6.71 | 66.09 | 3.59 | 5.68 | 71.95 | 6.97 | 9.95 | 66.03 | 1486.00 | 1404.00 | 35.88 |
> |  | MagR | **5.75** | **7.25** | 63.53 | **5.09** | **6.65** | **66.66** | 3.50 | 5.63 | 71.88 | 7.65 | 10.04 | 66.43 | 1856.19 | 1894.26 | 36.23 |
> |  | NeUQI | 5.80 | 7.26 | 63.36 | **5.09** | 6.67 | 66.23 | **3.47** | **5.62** | **72.33** | **6.67** | **9.41** | **67.41** | **4.90** | **10.00** | **67.57** |
> | W4G128 | GPTQ | 5.62 | 7.12 | **64.53** | 5.00 | 6.56 | 67.35 | 3.42 | **5.58** | 72.18 | 6.19 | 8.99 | 67.23 | 3.40 | 6.41 | 74.20 |
> |  | MagR | 5.69 | 7.15 | 63.75 | 5.03 | 6.59 | 67.36 | 3.46 | 5.61 | 72.00 | 6.88 | 9.74 | 67.98 | 3.75 | 7.33 | 73.80 |
> |  | NeUQI | **5.60** | **7.09** | 64.27 | **4.98** | 6.56 | **67.45** | **3.41** | **5.58** | **72.42** | **6.12** | **8.87** | **68.66** | **3.17** | **6.25** | **74.39** |
>
>
> > W2: Missing State-of-the-Art (SOTA): The paper does not provide direct comparisons with QuIP or QuIP#, which are widely considered the benchmarks for ultra-low bit quantization. A direct comparision with these methods or an experiment suggested by question 3 would be helpful.
>
> For QuIP, the transformation-based technique it employs is similar to the Hadamard Transformation, and NeUQI is orthogonal to QuIP. Therefore, Table 3 shows the combined effect of NeUQI and the Hadamard Transformation. By analogy, a similar complementary effect is expected when combining NeUQI with QuIP.
>
> For QuIP#, which adopts a vector quantization scheme different from the uniform quantization paradigm considered in our work, it falls outside the scope of our work and is therefore not included in our comparison.
>
> > Questions
>
> For Q1, the answer is generally no. NeUQI and transformation-based methods optimize different aspects: transformation-based methods smooth the weight distribution, while NeUQI considers the weight distribution to obtain optimal scale and zero-point parameters.
>
> For Q2, the benefit does not diminish. As shown in Appendix A, even when the weight distribution is perfectly uniform, the Min-Max formula does not yield the optimal scale. Therefore, as in Table 3, after applying the Hadamard transform, NeUQI can still further reduce quantization error.
>
> For Q3, as discussed above, NeUQI optimizes aspects that transformation-based methods cannot address, including those that cannot be improved even with fully smoothed weight distributions.

---

> > ### Author Rebuttal · Reviewer_WP2o · 2026-04-01
> >
> > Thank you for the detailed rebuttal and the additional experiments.
> >
> > I acknowledge that W1 has been satisfactorily addressed. The extended comparisons with MagR across W3 and W4 settings on multiple LLaMA families are appreciated and largely resolve my concern about baseline coverage. The results consistently show NeUQI's competitiveness or superiority over MagR, which strengthens the empirical case.
> >
> > The responses to Q1 and Q3 are also well-reasoned. The distinction that transformation-based methods and NeUQI optimize orthogonal aspects of the quantization problem is clear and convincing.
> >
> > However, I would like to request additional empirical support for the following two points before finalizing my recommendation:
> >
> > - W2 / Q2 (Complementarity with transformation-based methods): The authors claim, via analogy to Table 3, that a similar complementary gain is expected when combining NeUQI with QuIP. While the theoretical argument based on Appendix A is reasonable, an analogy is not a substitute for empirical evidence. A direct experiment combining NeUQI with QuIP would be necessary to fully substantiate this claim and address the concern that the benefit of near-optimal initialization diminishes after distribution smoothing.
> >
> > I am willing to raise my rating if the authors can provide this additional empirical evidence.

---

> > > ### Author Response · Authors · 2026-04-02
> > >
> > > We sincerely appreciate your assessment and are grateful for the time and effort you devoted to evaluating our paper. We are encouraged that our clarifications partially addressed your concerns. If accepted, we will ensure that all additional experiments introduced during the rebuttal are fully incorporated into the final version. We would also like to ask whether all of your concerns have now been addressed. If any issues remain unresolved or if you have further questions, we would be very happy to continue the discussion.
> > >
> > > The 2-bit quantization results for QuIP and QuIP combined with NeUQI are as follows:
> > > | Method | LLaMA 2 7B |  |  | LLaMA 2 13B |  |  | LLaMA 3 8B |  |  |
> > > |---|---|---|---|---|---|---|---|---|---|
> > > |  | WikiText2 | C4 | Acc | WikiText2 | C4 | Acc | WikiText2 | C4 | Acc |
> > > | QuIP | 149.30 | 110.38 | 36.19 | 13.18 | 14.31 | 49.41 | 150.27 | 121.28 | 35.56 |
> > > | QuIP + NeUQI | **15.24** | **16.98** | **50.04** | **8.63** | **10.34** | **56.93** | **33.07** | **33.33** | **41.87** |

---

### Decision · Program_Chairs · 2026-04-30

**Decision:**

Accept (regular)

**Comment:**

This paper studies initialization of scale and zero-point parameters for low-bit uniform PTQ and proposes NeUQI as a near-optimal alternative to the standard Min-Max rule. Reviewers found the problem practically important and the method technically sound, with consistent empirical gains across multiple LLaMA and Qwen models.

Initial concerns were mainly about baseline coverage and whether the proposed initialization would still provide benefit when combined with downstream fine-tuning or transformation-based approaches. In the rebuttal, the authors provided additional comparisons and ablations that addressed these points. Following the discussion, several reviewers indicated that their concerns were resolved and updated their assessment accordingly.

While some limitations remain (e.g., limited comparison with certain concurrent methods and the performance gap in extreme low-bit settings), the paper provides a useful and easy-to-integrate improvement for practical PTQ pipelines.

Based on the reviewer feedback and the clarifications provided during rebuttal, I recommend acceptance.